# Hybrid Latent Representations for PDE Emulation

**Ali Can Bekar**
Helmholtz Centre Hereon
Geesthacht, Germany
ali.bekar@hereon.de

**Siddhant Agarwal**
Helmholtz Centre Hereon
Geesthacht, Germany
siddhant.agarwal@hereon.de

**Christian Hüttig**
Institute of Space Research
German Aerospace Center (DLR)
Berlin, Germany
christian.huettig@dlr.de

**Nicola Tosi**
Institute of Space Research
German Aerospace Center (DLR)
Berlin, Germany
nicola.tosi@dlr.de

**David S. Greenberg**
Helmholtz Centre Hereon
Geesthacht, Germany
david.greenberg@hereon.de

## Abstract

For classical PDE solvers, adjusting the spatial resolution and time step offers a trade-off between speed and accuracy. Neural emulators often achieve better speed-accuracy trade-offs by operating on a compact representation of the PDE system. Coarsened PDE fields are a simple and effective representation, but cannot exploit fine spatial scales in the high-fidelity numerical solutions. Alternatively, unstructured latent representations provide efficient autoregressive rollouts, but cannot enforce local interactions or physical laws as inductive biases. To overcome these limitations, we introduce hybrid representations that augment coarsened PDE fields with spatially structured latent variables extracted from high-resolution inputs. Hybrid representations provide efficient rollouts, can be trained on a simple loss defined on coarsened PDE fields, and support hard physical constraints. When predicting fine- and coarse-scale features across multiple PDE emulation tasks, they outperform or match the speed-accuracy trade-offs of the best convolutional, attentional, Fourier neural operator-based and autoencoding baselines.

## 1 Introduction

Integrating partial differential equations (PDEs) can be computationally expensive, particularly for chaotic and turbulent systems with fine-scale structures requiring high spatiotemporal resolution. For traditional solvers, discarding fine-scale details reduces both cost and accuracy, but data-driven emulators can push the cost–accuracy frontier beyond conventional limits. State-of-the-art emulators [Kochkov et al., 2021, Stachenfeld et al., 2021, Li et al., 2021] employ supervised training on PDE fields coarsened by local averaging, a straightforward approach emphasizing speed and stability. But are local averages an optimal representation for emulation? Fine-scale information is crucial for convolutional network accuracy [Wang et al., 2020a], and emulators tend to neglect low-amplitude spatial frequencies [Rahaman et al., 2019, Lippe et al., 2023]. Fine-scale features could potentially reduce emulation errors, which tend to accumulate and amplify over autoregressive rollouts. The main alternative to coarsening is unstructured latent representations [Wiewel et al., 2020], but problems with scalability, generalization and physical consistency hinder widespread adoption.

39th Conference on Neural Information Processing Systems (NeurIPS 2025).

We introduce compact hybrid representations of fine-scale PDE fields, combining local averages with spatially structured latent variables. Hybrid representations exploit fine-scale information, allow efficient coarse-scale emulation and admit hard physical constraints [McGreivy and Hakim, 2023]. Hybrid representations provide more favorable cost-accuracy trade-offs than strong baselines on challenging emulation tasks, maintain accuracy at higher coarsening factors and benefit more from unrolled training. After time stepping, they can be used to reconstruct high-resolution PDE fields more accurately than super-resolution or high-resolution rollouts with baseline emulators.

## 2 PDE Emulation

For a 2D PDE with $m$ variable fields, the space- and time-discretized system state at time step $t$ is $X_t \in \mathbb{R}^{m \times H \times W}$. A *reference solver* $\mathcal{S}$ updates $X_t$, starting from prior-drawn initial conditions:

$$X_0 \sim p(X_0) \tag{1}$$
$$X_t = \mathcal{S}(X_{t-1}), \qquad 1 \leq t \leq T \tag{2}$$

We assume a fixed time step length $\delta$ between solver outputs. The reference solver runs slowly but accurately at the *reference resolution* $H \times W$ to generate training and testing data for emulation tasks. For some PDEs and solvers, $\mathcal{S}$ can be slow due to adaptive time steps that require many internal steps to produce outputs every $\delta$ time units, or because a system of equations must be solved iteratively at each time step.

**Coarsening** Let $\mathcal{A}_r : X_t \to x_t \in \mathbb{R}^{m \times H/r \times W/r}$ denote the coarsening operator reducing spatial resolution by a *coarsening factor* $r$ on each spatial axis. The nature of $\mathcal{A}_r$ depends on the PDE and discretization (appendix A, Fig. 6); we choose $\mathcal{A}_r$ to respect conservation laws where possible. Most emulation studies use coarsening in some way, but consider it a detail of data preparation. Since we compare different representations and resolutions deriving from the same reference simulation, we denote coarsening explicitly and consider it part of the emulator.

**Problem statement** We aim to predict coarsened future PDE fields quickly and accurately. We assume high resolution reference simulations $\{X_0, X_1, \ldots, X_T\}$ are available as training data, and that the coarsening operator $\mathcal{A}_r$ is known. A **PDE emulator** is a method that predicts future PDE fields from known initial conditions, with accuracy to be evaluated at a *target resolution* $h^* \times w^* = H/r^* \times W/r^*$ that may be coarser than the simulations ($r^* \geq 1$). That is, given $X_t$ we wish to predict $\mathcal{A}_{r^*}(X_{t+k})$. The training data, reference resolution $H \times W$, target resolution $h^* \times w^*$ and rollout length $k$ together define an **emulation task**.

**Accuracy Metrics** We measure correlation and mean-squared-error (MSE) for the PDE's $m$ fields. In some cases, we also compare the emulator's velocity and energy spectra to those of the reference solver, or compare the time evolution of the L1 norm of emulators' one-step updates to those of PDE fields at the target resolution [Kochkov et al., 2021, Kohl et al., 2024].

**Inference speed** Since comparison to reference solutions defines accuracy, only emulators with inference faster than $\mathcal{S}$ are useful. An emulator more accurate than alternatives with equal or faster inference speed is the best method *for the given speed and task*. We measure inference speed as the slope of wall-clock time as a function of $k$. We report additional details on timing overheads relevant for low $k$ in appendix B.

### 2.1 Fixed-resolution Neural Emulation

Most emulation studies trained fixed-resolution neural emulators (FRNEs) to operate autoregressively on coarsened PDE fields $x_t = \mathcal{A}_r(X_t)$. A loss defined on output fields is optimized, such as MSE:

$$\mathcal{L}_{\text{FRNE}} = \sum_{k=1}^{K} \left\| \mathcal{M}^{(k)}(x_t) - x_{t+k} \right\|_2^2 = \sum_{k=1}^{K} \left\| \mathcal{M}^{(k)} \circ \mathcal{A}_r(X_t) - \mathcal{A}_r(X_{t+k}) \right\|_2^2 \tag{3}$$

$\mathcal{M}^{(k)}$ is emulator autoregressively applied to the input fields $k$ times. $K = 1$ trains on single-step predictions, while $K > 1$ uses autoregressive rollouts. The *rollout resolution* $h \times w = H/r \times W/r$ is used by the emulator to advance the coarsened PDE fields in time.

A common choice is matching the rollout and target resolutions ($r = r^*$), but in general $r$ is an adjustable hyperparameter, and together with $\mathcal{M}$'s neural architecture can be chosen to emphasize

accuracy or speed. $r = 1$ trains at the reference resolution, but for most tasks and architectures, this results in inference slower than $\mathcal{S}$ [McGreivy and Hakim, 2024]. Instead, FRNEs for chaotic and turbulent PDEs have been applied most effectively with $4 \leq r \leq 64$ [Zhou et al., 2024a, Stachenfeld et al., 2021, Gupta and Brandstetter, 2023, Kohl et al., 2024, Zhang et al., 2024, Wang and Wang, 2024, Wang et al., 2020b]. While our focus here is on PDEs defined on regular grids, this approach has also been successful on irregular meshes [Brandstetter et al., 2022a] or for weather forecasting [Lam et al., 2023].

FRNEs exploit neural networks' better tolerance for coarsening than numerical solvers, and have employed a wide range of architectures incorporating convolutional, Fourier, attention or graph-based layers [Gupta and Brandstetter, 2023, Li et al., 2021, Hao et al., 2024, Brandstetter et al., 2023, Horie and Mitsume, 2024, 2022, Sanchez-Gonzalez et al., 2020]. Because the learned time stepping operator $\mathcal{M}$ operates on coarsened PDE fields, it can be constructed to impose physical constraints such as incompressibility [Wandel et al., 2021, Wiewel et al., 2020] or conservation laws [Watt-Meyer et al., 2025, Verma et al., 2024, Horie and Mitsume, 2024]. Ideally, $\mathcal{M}$ should be faster than $\mathcal{S}$ but only slightly less accurate, and more accurate than decreasing the reference solver's resolution to obtain the same inference speed. A disadvantage of FRNEs is that $\mathcal{A}_r$ destroys information, so $X_t$ may contain fine-scale features useful for predicting $x_{t+k}$ that are unavailable in $x_t$.

## 2.2 Unstructured Latent Space Emulation

Unstructured Latent Space Emulators (ULSEs) learn time stepping for a compact, unstructured latent representation of PDE fields. They are commonly trained in an encoder-processor-decoder framework: PDE fields coarsened to an *encoding resolution $H/r \times W/r$* are encoded to a vector $z_t$, autoregressively rolled out, and decoded back to PDE fields at the encoding resolution. This results in the loss function:

$$\mathcal{L}_{ULSE} = \left\| \mathcal{D} \circ \mathcal{M}_z^{(K)} \circ \mathcal{E}(x_t) - x_{t+K} \right\|_2^2 = \left\| \mathcal{D} \circ \mathcal{M}_z^{(K)} \circ \mathcal{E} \circ \mathcal{A}_r(X_t) - \mathcal{A}_r(X_{t+K}) \right\|_2^2 \quad (4)$$

Where $\mathcal{E}$ is the encoder, $\mathcal{D}$ is the decoder and $\mathcal{M}_z$ is the propagator operating on latent representations. $\mathcal{L}_{ULSE}$ is efficient since it is only calculated once after an efficient $K$-step rollout in latent space. For $r = 1$, the encoder operates directly on reference PDE fields, and can learn to exploit any features of $X_t$, but in most studies $r > 1$ [Hao et al., 2024, Li et al., 2023, Serrano et al., 2024, Alkin et al., 2024, Wu et al., 2022, Knigge et al., 2024, Wang and Wang, 2024].

Alternatively, $\mathcal{E}$ and $\mathcal{D}$ can first be trained as an autoencoder, and then time stepping can be learned using fixed latent vectors $z_t$:

$$\mathcal{L}_z = \sum_{k=1}^{K} \left\| \mathcal{M}_z^{(k)}(z_t) - z_{t+k} \right\|_2^2 = \sum_{k=1}^{K} \left\| \mathcal{M}_z^{(k)} \circ \mathcal{E} \circ \mathcal{A}_r(X_t) - \mathcal{E} \circ \mathcal{A}_r(X_{t+k}) \right\|_2^2 \quad (5)$$

$\mathcal{L}_z$ efficiently computes a loss at each time step without involving PDE fields, but appropriately weighting the dimensions of $z$ is challenging. Fluid dynamics have been emulated using $\mathcal{L}_z$ [Wiewel et al., 2019, HAN et al., 2022, Yin et al., 2023, Li et al., 2025], albeit with few competitive results on challenging benchmark tasks.

ULSEs provide fast training and inference with long rollouts, as their compact latent spaces lack spatial structure. However, ULSEs lack the spatial inductive biases that improve performance and robustness in many computer vision tasks, and $x_t$ and $x_{t+1}$ might not be close in the latent space. Furthermore, constructing $\mathcal{M}_z$ to obey physical constraints defined for $x_t$ is usually not possible.

## 3 Emulation with Hybrid Representations

We introduce hybrid representations (Fig. 1a), which extend coarsened PDE fields with additional channels of learned latent variables to combine the advantages of FRNEs and ULSEs. Like the coarsened PDE fields used by FRNEs, they are spatially structured and support physical constraints on the learned time stepping operator. Like ULSE latent variables, they extract information with a trained encoder instead of relying solely on $\mathcal{A}_r$, before rolling out cost-effectively. Emulation with hybrid representations jointly trains an encoder and processor as in eq. 4, but does so using an efficient loss that has more in common with eq. 3 and 5. We first describe this strategy in full generality, reserving architectural details for sec. 4.

**Encoder** Hybrid representations consist of coarsened PDE fields $x_t \in \mathbb{R}^{m \times h \times w}$, together with $n$ latent variables fields $c_t \in \mathbb{R}^{n \times h \times w}$. $x_t$ is computed by $\mathcal{A}_r$ and $c_t$ by an encoder $\mathcal{E}_c$ applied to $\mathcal{A}_{r'}(X_t)$, with $1 \leq r' < r$.

$$(x_t, c_t) = (\mathcal{A}_r(X_t), \mathcal{E}_c \circ \mathcal{A}_{r'}(X_t)) = \mathcal{E}_{\text{HR}}(X_t) \tag{6}$$

$\mathcal{E}_c$ is trained to extract latent variables from $\mathcal{A}_{r'}(X_t)$ while maintaining spatial structure. Its encoding resolution $H/r' \times W/r'$ is higher than its rollout resolution $H/r \times W/r$.

**Processor** A second network $\mathcal{M}_{\text{HR}}$ carries out time stepping on the hybrid representation:

$$(\tilde{x}_{t+1}, \tilde{c}_{t+1}) = \mathcal{M}_{\text{HR}}(x_t, c_t) = (\mathcal{M}_x(x_t, c_t), \mathcal{M}_c(x_t, c_t)) \tag{7}$$

$\mathcal{M}_{\text{HR}}$ computes autoregressive rollouts entirely within the hybrid representation at the rollout resolution $H/r \times W/r$. As the representation advances in time, coarsened PDE fields and latent variables mutually influence each other. The notation $\tilde{x}_t, \tilde{c}_t$ marks these variables as having been time-stepped by the processor, as opposed to $x_t$ and $c_t$ which are computed by $\mathcal{A}_r$ and $\mathcal{E}_c \circ \mathcal{A}_{r'}$ respectively.

**Loss** We wish to avoid costly loss calculations at the encoding resolution (eq. 4) or over the numerous, non-physical channels of $c_t$ (eq. 5). We therefore jointly train all encoder and processor parameters by minimizing prediction errors only over coarsened PDE fields $x_t$ at the lower rollout resolution:

$$\mathcal{L}_{\text{HR}} = \sum_{k=1}^{K} \left\| \mathcal{M}_x \circ \mathcal{M}_{\text{HR}}^{(k-1)} \circ \mathcal{E}_{\text{HR}}(X_t) - x_{t+k} \right\|_2^2 \tag{8}$$

While $\mathcal{L}_{\text{HR}}$ does not include error terms on $c_t$, minimizing it still encourages $\mathcal{E}_c$ and $\mathcal{M}_{\text{HR}}$ to generate latent variables that improve prediction of future states $x_{t+k}$, especially for a long training rollout length $K$. We reasoned that while $c_t$ ought to improve prediction of $x_{t+k}$, it would be unrealistic to expect the processor to accurately predict $c_{t+k}$ without access to $X_{t+k}$. We found this simple and efficient loss more effective in practice than alternatives involving error terms on $c_{t+k}$ or $X_{t+k}$. Since $\mathcal{L}_{\text{HR}}$ only includes error terms for variables with the same units as the original PDE fields, it can use data normalizations independent of the training process and architecture.

**Physical Constraints** An advantage of explicitly including $x_t$ in hybrid representations is that $\mathcal{M}_x$ can be constructed to respect physical constraints. For example, for a conserved PDE field such as energy or density, we can learn fluxes between neighboring grid cells [LeVeque, 2002]. Similarly, for a pair of channels $(u_t, v_t) \subset x_t$ representing the velocity of an incompressible fluid, we can learn a vector potential $a_{t+1}$ whose curl yields the desired divergence-free field $(\tilde{u}_{t+1}, \tilde{v}_{t+1}) = \nabla \times a_{t+1}$ [Wandel et al., 2021]. Hybrid representations allow these physical principles to be enforced as hard constraints, while nonetheless improving the time stepping of $x_t$ with additional information from $c_t$. Together with a physically consistent local averaging operator $\mathcal{E}_x$ (Fig. 6), this provides end-to-end physical consistency from $X_t$ to $\tilde{x}_{t+k}$.

**Decoder** When the rollout resolution exceeds the target resolution, we can predict $\mathcal{A}_{r^*}(X_{t+k})$ by coarsening $\tilde{x}_{t+k}$. Otherwise, we train a decoder $\mathcal{D}_{\text{HR}}$ to reconstruct $X_t^* \approx \mathcal{A}_{r^*}(X_t)$, using a fixed interpolation operator $\mathcal{D}_x$ (Fig. 7) and learned correction $\mathcal{B}_c$:

$$X_t^* = \mathcal{D}_{\text{HR}}(\tilde{x}_t, \tilde{c}_t) = \mathcal{D}_x(\tilde{x}_t) + \mathcal{B}_c(\tilde{x}_t, \tilde{c}_t) \tag{9}$$

$\mathcal{D}_x$ and the output layer of $\mathcal{B}_c$ are chosen in a PDE-specific way to enforce conservation laws and/or incompressibility as hard constraints (appendix A.3).

To train $\mathcal{D}_{\text{HR}}$ we minimize:

$$\mathcal{L}_{\mathcal{D}} = \sum_{k=1}^{K} \left\| \mathcal{A}_{r^*}(X_{t+k}) - X_{t+k}^* \right\|_2^2 = \sum_{k=1}^{K} \left\| \mathcal{A}_{r^*}(X_{t+k}) - \mathcal{D}_{\text{HR}} \circ \mathcal{M}_{\text{HR}}^{(k)} \circ \mathcal{E}_{\text{HR}}(X_t) \right\|_2^2 \tag{10}$$

For efficiency, and so $c_t$ emphasizes accurate rollouts of $\tilde{x}_{t+k}$ over decoding of $X_{t+k}^*$, we train $\mathcal{D}_{\text{HR}}$ with the encoder and processor frozen after training. Thus, $\mathcal{D}_{\text{HR}}$ is trained on triplets $(\tilde{x}_t, \tilde{c}_t, X_t)$ without backpropagation through time, and is not used to train the encoder and processor (eq. 8).

While standard super-resolution maps $\tilde{x}_t \to X_t^*$ and a ULSE decoder maps $z_t \to X_t^*$, our decoder $\mathcal{D}_{\text{HR}}$ takes both $\tilde{x}_t$ and $\tilde{c}_t$ as inputs. $\tilde{c}_t$ provides information unavailable in $\tilde{x}_t$, while the presence of $\tilde{x}_t$ allows physical constraints to be imposed.

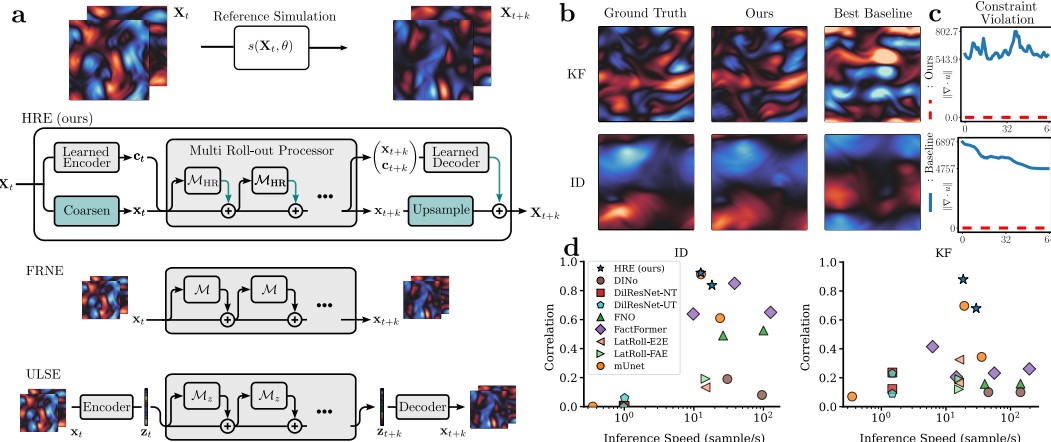

Figure 1: (**a**) Top: hybrid representation emulators combine locally averaged PDE fields with latent variables. A processor network evolves both coarsened and latent fields in time, and an optional decoder reconstructs high-resolution fields. Teal arrows and boxes respect conservation laws and incompressibility. Middle: fixed-resolution neural emulators. Bottom: unstructured compact latent space emulators. (**b**) Reference solver vs. HREs and mUnet baseline (with super-resolution) with rollouts at $64^2$ for KF and $32^2$ for ID and encoding from $512^2$. KF is shown after 8.5 and ID after 7.75 seconds. (**c**) Violation of incompressibility for HREs and mUnet. (**d**) Correlation with reference solver vs. inference speed for HREs and baselines. Multiple points on each cost-accuracy frontiers were obtained by adjusting the resolution used to advance rollouts (HREs/FRNEs) or to provide encoder input (ULSEs).

## 4  Emulation Architectures

Here we describe architectural details for baseline emulation approaches operating on coarsened and high-resolution PDE fields, alongside our hybrid representation-based emulators. FRNEs and HREs learn additive updates estimating $x_t - x_{t-1}$. We did not seek to match parameter counts (Table 3) across emulators, instead choosing one or more hyperparameter settings for each emulator to obtain overlapping ranges of inference speeds and facilitate comparison of cost-accuracy trade-offs across architectures. Full architectural details appear in appendix A.

**Modern U-Net with Attention (mUnet)** the U-Net architecture [Ronneberger et al., 2015] has been optimized for use as an FRNE, and performed well in many emulation tasks and benchmark studies [Gupta and Brandstetter, 2023, Kohl et al., 2024]. We also examined the **SineNet** variant [Zhang et al., 2024], but found it to have a less favorable cost-accuracy trade-off (Fig. 16).

**FactFormer** [Li et al., 2023] is a Transformer [Vaswani et al., 2017] designed for operator learning. It uses axial factorized attention to learn integral operators while maintaining model scalability. We train FactFormer as a ULSE at encoding resolutions $32^2, 64^2, 256^2$ and $512^2$.

**DPOT** [Hao et al., 2024] is an attention-based FRNE, trained on diverse datasets and tasks as a foundation model using 1-step prediction loss and input noise. It aggregates coarsened fields from multiple input time steps, and its Fourier attention layers use an MLP for parameter-efficient approximation of frequency space computations. DPOT-S (31M parameters, matching our HREs) was fine-tuned in few-shot mode at resolutions $32^2$ and $64^2$ on each task before evaluation.

**DINo** [Yin et al., 2023] is a ULSE combining implicit neural representations [Sitzmann et al., 2020, Tancik et al., 2020] with neural ODEs [Chen et al., 2018]. Input fields are used to compute an unstructured latent vector through an auto-decoding framework, propagated through time by a neural ODE and decoded to predict PDE fields. DINo can produce outputs anywhere along a continuous time axis, predict PDE fields from incomplete initial conditions and generalize across resolutions.

**Dilated ResNet (DilResNet)** [Stachenfeld et al., 2021] is an FRNE designed for turbulent flows using dilated convolutions. We train it using either input noise (DilResNet-NT) or unrolled training (DilResNet-UT). In the main text, we use DilResNet-NT and refer to it simply as DilResNet.

**Fourier Neural Operator (FNO)** [Li et al., 2021] is an FRNE that combines filtering in Fourier space with local linear operations, and was originally introduced for PDE emulation.

**Hybrid Representation Emulators (HREs, ours)** employ encoder, processor and decoder modules (Fig. 1, sec. 3). Their design closely follows mUnet, our strongest baseline, to evaluate specific benefits of hybrid representations instead of architectural differences. The encoder $\mathcal{E}_c$ uses $\log_2(r/r')$ mUnet downsampling blocks to reach the desired coarsening factor. The processor is an mUnet with $n$ extra input/output channels. Except where otherwise stated, we use $n = 16$ learned latent variable fields. The decoder corrects interpolation $\mathcal{D}_x$ using a physics-constrained additive update (eq. 14, 16) from a network $B_\psi$ constructed from mUnet upsampling blocks. The encoder, processor and decoder use flux/vector potential outputs to impose physical constraints.

**LatRoll** We implemented an HRE-like baseline with only learned variables in its latent representation. This model used the same encoder to calculate $c_t$, the same mUnet processor to compute latent space rollouts, the same decoder to compute $X_{t+k}^*$ from $\tilde{c}_{t+k}$ and the same physical constraints but did not include coarsened PDE fields $x_t$ as input to the processor or decoder. We trained LatRoll using a loss on $X_t$, obtaining best results when first training and freezing an autoencoder (LatRoll-FAE). We also report results for end-to-end training (LatRoll-E2E).

**Super-resolution** To evaluate baselines at higher target resolutions than their outputs, we trained super-resolution networks [Harder et al., 2023, Stengel et al., 2020, Esmaeilzadeh et al., 2020]. These took only $x_t$ as input and their outputs directly estimated $X_t$, but their architectures were otherwise identical to the $B_\psi$ of HREs, and were not trained jointly with emulators (Table 3).

## 5 Training

**Loss and Optimization** We train baselines as described in the respective studies. We standardize reference fields $X_t$. We normalize both components of each velocity field jointly, so the momentum conservation and incompressibility enforced by flux- and vector potential-based output layers are not violated. We use the AdamW optimizer [Loshchilov and Hutter, 2019] except on DINo which uses Adam [Kingma and Ba, 2015]. A cosine scheduler decays learning rate from $10^{-4}$ to $10^{-6}$ [Loshchilov and Hutter, 2017] for our HREs, mUnet, FNO and DilResNet, while DINo and FactFormer used their original schedulers. Further training details appear in appendix A.1.

**Unrolled Training and Input Noise** We train mUnet, FNO, and hybrid representations on a rollout curriculum of lengths $K = [1, 2, 4, 8, 16]$, each for 100 epochs with early stopping and patience of 75 epochs. We did not observe benefits from unrolled training with DilResNet, but instead added Gaussian noise ($\mu = 0$, $\sigma = 10^{-4}$) to coarse inputs $x_t$ as previously described [Stachenfeld et al., 2021]. For DINo and FactFormer, we follow the original papers' training schedules.

## 6 Datasets

We generated 5 datasets with numerical solvers chosen to respect physical laws of the integrated PDEs. Each dataset employs a fixed time step $\delta$ longer than the solver's internal time step; we choose $\delta$ based on previous studies where possible (details in appendix C). Table 10 lists all model-dataset combinations we employed.

**2D Kolmogorov Flow (KF)** consists of solutions to the 2D incompressible Navier-Stokes equations:

$$\partial_t \boldsymbol{u} + \nabla \cdot (\boldsymbol{u} \otimes \boldsymbol{u}) - \frac{1}{\mathrm{Re}} \Delta \boldsymbol{u} + \frac{1}{\rho} \nabla p - \boldsymbol{f} = \boldsymbol{0} \qquad\qquad \nabla \cdot \boldsymbol{u} = 0 \qquad (11)$$

for velocity $\boldsymbol{u}$, constant density $\rho$, pressure $p$, forcing $\boldsymbol{f}$ and Reynolds number Re. We set $(\rho, \mathrm{Re}, \nu) = (1, 10^3, 10^{-3})$ and $\boldsymbol{f} = \sin(4y)\hat{\boldsymbol{e}}_1 - 0.1\boldsymbol{u}$, resulting in statistically stationary turbulent flow.

**2D Kolmogorov Flow with $\mathrm{Re} = 4000$ (KFHR)** As the smallest eddies scale with $1/\sqrt{\mathrm{Re}}$, we quadruple the Re to halve eddy size. This produces finer-scale structures in high-resolution fields.

**2D Incompressible Decaying Turbulence (ID)** has the same dynamics as KF but no forcing, resulting in transient rather than stationary dynamics. Larger structures develop as small eddies dissipate.

**2D Compressible Decaying Turbulence (CD)** solves the compressible Navier-Stokes equations:

$$\partial_t \rho + \nabla \cdot (\rho \boldsymbol{v}) = 0 \qquad (12a)$$

$$\partial_t (\rho \boldsymbol{v}) + \nabla \cdot (\rho \boldsymbol{v} \otimes \boldsymbol{v}) = -\nabla p + \nabla \cdot \boldsymbol{\tau} \qquad (12b)$$

$$\partial_t E + \nabla \cdot [(E + p)\boldsymbol{v} - \boldsymbol{v} \cdot \boldsymbol{\tau}] = 0 \qquad (12c)$$

for density $\rho$, velocity $\boldsymbol{v}$, pressure $p$, viscous stress tensor $\boldsymbol{\tau}$ and total energy $E = \epsilon + (\rho v^2)/2$. We use $(\zeta, \eta, M) = (10^{-8}, 10^{-8}, 0.4)$, with $M$ the initial Mach number, resulting in decaying turbulence as small eddies coalesce through an energy cascade. This dataset is challenging due to a CFL (Courant-Friedrichs-Levy) condition of 60, and nonlocal behavior of density and energy fields and the presence of shockwaves.

**1D Kuramoto-Sivashinsky Equation (KS)** models chaotic stationary dynamics of a scalar field $u$:

$$\partial_t u + \partial_x(u^2/2 + \partial_x u + \nu \partial_{xxx} u) = 0 \tag{13}$$

Nonlinear advection transfers energy between low and high energy modes. KS is a common benchmark task for neural emulators [Stachenfeld et al., 2021, Lippe et al., 2023, Schiff et al., 2024].

## 7    Experiments

**Cost vs. Accuracy of Neural Emulators**    We first evaluated HREs and baselines on emulation tasks at target resolution $32^2$ ($r^* = 64$). Where possible, we trained emulators at multiple rollout and encoding resolutions to obtain multiple points on their cost-accuracy curves (Fig. 1d). For all tasks, at $64^2$ rollout resolution HRE rollouts were equally or more correlated to reference simulations than all baselines. Similarly, HREs with $32^2$ encoding resolution were more accurate than all baselines except mUnet at $64 \times 64$ rollout resolution, which was slower. Only mUnet matched or approached the accuracy of HREs on any task, though DilResNet was sometimes comparable for shorter rollouts. When comparing methods at a common rollout or encoding resolution of $32^2$, the gap between HRE and baselines widened: HREs had 30-50% lower RMSE after 64 time steps (Table 1). Since hybrid representations use a processor architecture nearly identical to mUnet, they achieve a nearly identical inference speed. All emulators were faster than numerical simulations, and FactFormer and FNO were the fastest but considerably less accurate than HREs and slightly less accurate than mUnet on almost all tasks. A comprehensive quantitative comparison is provided in Tables 5-6.

HREs showed the greatest benefit over baselines on the KF task (Fig. 2-left, 9-8), where they exhibited higher correlation and lower MSE than mUnet by a wide margin across 64-step rollouts at both resolutions. HREs also tracked the average $\ell_1$ norm velocity tendencies more closely than other methods (Fig. 2-bottom left). For the ID and KFHR tasks, HRE was more accurate at $32^2$ rollout resolution but comparable to mUnet at $64^2$. For the challenging CD task we analyzed only up to 32-step rollouts at inference time, revealing that HREs were more accurate than other baselines at both resolutions (Fig. 11-12). However, the improvement from using hybrid representations was less prominent compared to ID and KF cases. The weaker performance of HREs on CD may arise from discontinuities in the density and energy fields, making the extracted fine-scale information from high-resolution inputs harmful for longer rollouts. Moreover, the dataset's high CFL number and the limited receptive field of the propagators could exacerbate instabilities. Notably, DilResNet and FNO's non-local nature might be the reason for their ability to match the performance of mUnet on this task.

We further compared the frequency spectra of TKE fields for both reference simulations and emulators (Fig. 3), following previous work [Kochkov et al., 2021] by normalizing with the fifth power of the wavenumber. After 64-step rollouts, hybrid representations produced spectra closer to the reference simulations than baselines relying solely on coarsened PDE fields. This improvement suggests that information from high spatial frequencies encoded in the $c_t$ fields (Figs. 13-14) enables the low-resolution emulator to capture the higher-frequency components of the TKE fields more accurately (additional results in Figs. 18-19).

Table 1: Rollout performance across emulators for KF and ID experiments at rollout resolution $32^2$ (encoding resolution for DINo and FactFormer). We report MSE and correlation (Corr.) of field variable $u$ after 64 rollout steps. Best model performance in **bold** and second underlined.

|  | KF | | ID | |
|---|---|---|---|---|
|  | RMSE $\downarrow$ | Corr. $\uparrow$ | RMSE | Corr. |
| OURS | **0.71**$_{\pm.23}$ | **0.68**$_{\pm.21}$ | **0.29**$_{\pm.08}$ | **0.96**$_{\pm.03}$ |
| MUNET | 1.07$_{\pm.20}$ | 0.35$_{\pm.21}$ | 0.58$_{\pm.34}$ | 0.78$_{\pm.25}$ |
| FNO | 1.29$_{\pm.21}$ | 0.12$_{\pm.19}$ | 0.71$_{\pm.27}$ | 0.74$_{\pm.20}$ |
| DILRESNET | 1.16$_{\pm.20}$ | 0.26$_{\pm.20}$ | 1.77$_{\pm.72}$ | 0.50$_{\pm.17}$ |
| FACTFORMER | 1.12$_{\pm.21}$ | 0.28$_{\pm.20}$ | 0.58$_{\pm.25}$ | 0.79$_{\pm.51}$ |
| DINO | 1.00$_{\pm.10}$ | 0.20$_{\pm.17}$ | 1.10$_{\pm.27}$ | 0.30$_{\pm.30}$ |
| DPOT-S | 1.85$_{\pm.24}$ | 0.20$_{\pm.17}$ | 1.65$_{\pm.64}$ | 0.18$_{\pm.24}$ |
| DILRESNET-UT | 1.19$_{\pm.22}$ | 0.26$_{\pm.22}$ | 0.84$_{\pm.19}$ | 0.72$_{\pm.16}$ |
| LATROLL-32-FAE | 1.15$_{\pm.17}$ | 0.14$_{\pm.16}$ | 0.96$_{\pm.22}$ | 0.39$_{\pm.35}$ |
| LATROLL-32-E2E | 1.23$_{\pm.17}$ | 0.20$_{\pm.16}$ | 1.20$_{\pm.38}$ | 0.27$_{\pm.43}$ |

To test generalization to data outside the training distributions, we compared HREs and mUnet on the IC task with the initial peak wave

number changed from 4.2 to 3, 5 or 6. As expected, performance on out-of-distribution (OOD) inputs worsened as peak wave number increased, but HREs continued to outperform mUnet in every case (Table 9). Thus, the representation learned by the HRE encoder does not become a liability when encountering OOD data.

**The Role of the Coarsening Ratio**  HREs retain information in $c_t$ that FRNEs at the same rollout resolution lack, so does their performance gap widen as $r$ increases? KF and ID experiments at rollout resolutions $32^2$ and $64^2$ confirm this ($r = 16$ and $8$, Fig. 8-9, 11-12), and in general HREs were better suited to large $r$ than baselines (Fig. 1d). FNOs showed the smallest accuracy decreases when doubling $r$, but were among the least accurate FRNEs overall. For KF, HREs outperformed mUnets at both resolutions, but for ID did so only at $32^2$, while for $64^2$ their performance was similar. To summarize, we observe that: (i) latent fields allow HREs to cope with higher $r$ increases, (ii) HREs outperform FRNEs only for sufficiently large $r$ (emphasizing speed over accuracy), and (iii) the minimum $r$ for which HREs beat FRNEs varies across tasks.

Since HREs benefit from latent features extracted from high-resolution PDE fields, why not simply train FRNEs on high resolution data? To test this, we trained mUnet with rollout resolution $512^2$, resulting in inference $> 30$ times slower than for rollouts at $64^2$. Somewhat counterintuitively, mUnet rollouts at $512^2$ were less accurate than at $32^2$ or $64^2$ rollouts, despite access to additional information from $X_t$ (Fig. 1d). One possible reason is that the physical size of receptive fields for a fixed mUnet architecture shrinks as $r$ decreases. Training on high resolution data also emphasizes fine scales not reflected in accuracy measures at the target resolution. In contrast, HREs work efficiently across resolutions by encoding $512^2$ PDE fields to predict future fields at $32^2$.

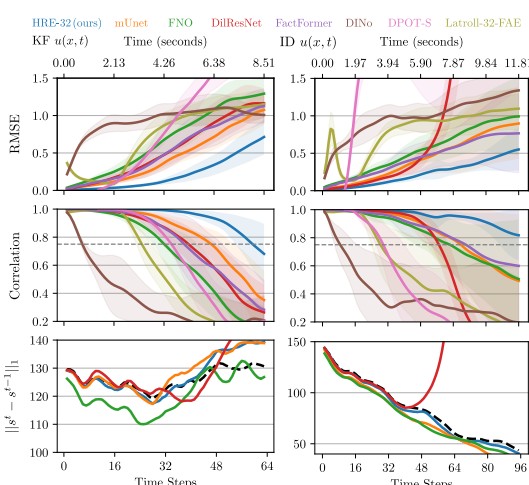

To test whether the trend would continue at still higher coarsening ratio, we trained HREs and mUnet on data from the KS task with a reference resolution of 2048 and rollout resolution 32 ($r = 64$). In contrast to other tasks for which HREs used an encoding resolution of $512^2$, here we

Figure 2: Emulator accuracy on KF and ID tasks (testing data, 16 ICs) at $32^2$ rollout/encoding resolution. Mean $\pm$ s.d. over ICs of RMSE (**top**) and correlation (**middle**), and tendency $\ell_1$ norm (**bottom**) vs. rollout length.

used no coarsening before the encoder ($r' = 1$). HREs outperformed mUnet by a wider margin than in previous experiments, achieving near-perfect predictions after 16 time steps in this challenging chaotic system, while mUnet's correlation dropped to 0.5 over the same rollout length (Fig. 15).

**Benefits of Unrolled Training**  Consistent with previous studies [List et al., 2024, Lam et al., 2023, Kohl et al., 2024], we observed improved rollout performance for emulators trained autoregressively. However, while accuracy for 64-step rollouts plateaued as a function of training rollout length for mUnet, hybrid representations continued to benefit from training on curricula of longer sequences, up to at least 16 steps (Fig. 4). When training on shorter rollout lengths, mUnet exhibits superior performance, suggesting that the encoder $\mathcal{E}_c$ requires long-rollout training to learn features in $c_t$ useful for longer prediction horizons.

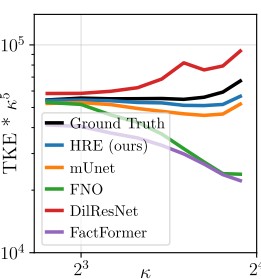

**Latent Space Dimensionality**  We investigated how the dimensionality $n$ of $c_t$ affects HRE accuracy on KF and ID tasks. $n$ too small could lose useful information from $\mathcal{A}_{r'}(X_t)$, while $n$ too large could overfit or produce unstable rollouts; we tested $n = 4$ and $n = 16$ (Table 7). For KF

Figure 3: Scaled TKE values for high spatial frequencies on KF ($32^2$).

at $64^2$ rollout resolution, reducing $n$ from 16 to 4 roughly doubled RMSE and decreased correlation by $2/3$. However, KF at $32^2$ rollout resolution was similar for $n = 4$ and 16. For ID $n = 16$ gave

RMSE 3 times lower, but had a smaller and less consistent effect on correlation (Fig. 8). $n$ negligibly impacted inference speed.

**Prediction at High Resolution** To evaluate PDE emulation of high-resolution input fields, we computed accuracy for KF and ID tasks at target resolution $512^2$ ($r^* = 4$, Fig. 21-22). Our HRE used encoding resolution $512^2$ ($r' = 4$), rollout resolution $64^2$ for KF ($r = 32$) and $32^2$ for ID ($r = 32$), and a physics-constrained decoder. As baselines, we considered mUnet at $r = 32$ and $r = 64$ followed by learned super-resolution (essentially the same decoder with only $x_t$ as an input), mUnet with $r = 4$ and FactFormer with $r = 4$ and $r = 8$. HRE outperformed all baselines at all rollout lengths on both tasks (Fig. 5, 20). Consistent with previous results at $r^* = 64$, mUnet with super-resolution consistently outperformed mUnet trained on $512^2$ for all but the shortest rollouts, and mUnet with $r = 4$ completely decorrelated with the reference solution before 64 steps. Both Fact-Former models performed slightly better than mUnet with $r = 4$.

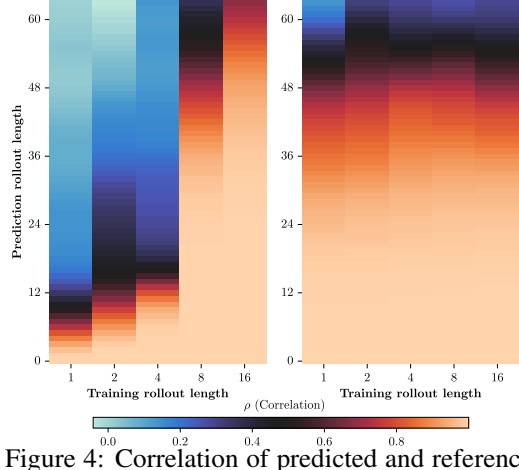

Figure 4: Correlation of predicted and reference PDE fields vs. rollout length during training and prediction on KF ($32^2$). Our HREs benefit from longer training rollouts, and achieve higher accuracy, while mUnet plateaus after 8-step training.

**Ablations** To identify benefits of hard physical constraints and learned latent fields for HREs, we trained a physics-constrained mUnet with no $c_t$, and an HRE without physical constraints (Fig. 17, Table 8). For 64-step rollouts, both models surpassed mUnet in accuracy but fell short of HREs. Unconstrained HREs yielded RMSE and correlation values only $\approx 20\%$ worse and $\approx 12\%$ lower respectively, but the physics-constrained mUnet had RMSE nearly $\approx 40\%$ higher and correlation $\approx 30\%$ lower. Despite its relatively strong quantitative performance, the unconstrained hybrid model produced oscillatory tendency norms and led to oscillatory behavior in the predicted $c_t$ fields during propagation (Fig. 17). This suggests that physical constraints can regularize latent representations.

# 8 Related Work

**Learning for Long Rollouts** A key challenge for neural PDE emulation is error accumulation over autoregressive rollouts, causing distribution shift for network inputs, instability or high gradient variance. To mitigate this, unrolled training and noise-injection strategies [List et al., 2024, Metz et al., 2021, Mikhaeil et al., 2022, Stachenfeld et al., 2021, Sanchez-Gonzalez et al., 2020, Kochkov et al., 2021, Lam et al., 2023, Hao et al., 2024] expose networks to their own outputs or corrupted inputs to enhance robustness. The pushforward trick [Brandstetter et al., 2022a, List et al., 2025] trains with rollouts but no backpropagation over time.

**Multiple Coarse Timesteps** we use $c_t$ to augment $x_t$, but other studies provide multiple coarsened states $x_{t-i}$ as additional inputs [Wang, 2021, Li et al., 2023] or via specialized aggregation layers [Buitrago et al., 2025, Hao et al., 2024] to predict $x_{t+1}$. This can improve performance, but incurs a computational cost and

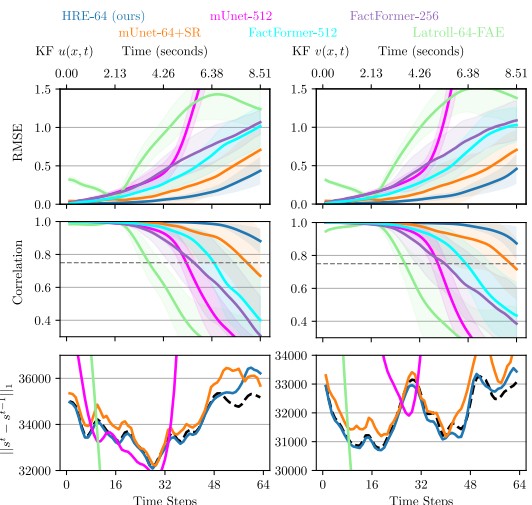

Figure 5: Accuracy at $512^2$ target resolution.

cannot learn which features to extract from $X_t$. Temporal bundling [Brandstetter et al., 2022a, Buitrago et al., 2025] instead predicts several output steps at once.

**Physics-derived Losses** minimizing a known PDE's residuals [Wandel et al., 2021, Tompson et al., 2017, Raissi et al., 2019] removes or reduces the need for reference simulations, but does not allow spatiotemporal coarsening for efficiency. **Hybrid Solvers** use machine learning to accelerate classical solvers or partial-physics models, or to correct discretization errors [Kochkov et al., 2021, Um et al., 2020, Pathak et al., 2020, Tompson et al., 2017, Paliard et al., 2022, Bar-Sinai et al., 2019].

**Neural Operators** emulate PDEs with autoregressive mappings between function spaces, allowing adaptation to different resolutions and geometries [Li et al., 2021, Alkin et al., 2024, Bonev et al., 2023, Lu et al., 2021]. [Hagnberger et al., 2024] take a similar approach to the time axis, encoding initial conditions and decoding at desired output times non-autoregressively. Likewise, **foundation models** use pretraining across diverse data and tasks to enable few- or zero-shot inference [Herde et al., 2024, Hao et al., 2024].

**Climate and Weather Prediction** drive PDE emulation research, as large-scale turbulence and chaos demand high spatiotemporal resolution, large domains and physical coherence [Kochkov et al., 2024, Keisler, 2022, Verma et al., 2024, Lam et al., 2023, Watt-Meyer et al., 2023, 2025, Price et al., 2024, Nguyen et al., 2023]. Some emulators enforce conservation laws and symmetries [McGreivy and Hakim, 2023, Verma et al., 2024, Wandel et al., 2021, Beucler et al., 2021, Lee and Carlberg, 2021, Yuval and O'Gorman, 2020, Lee and Carlberg, 2021, Brandstetter et al., 2022b, 2023].

## 9 Discussion and Outlook

By storing and propagating both coarsened and learned latent fields, HREs outperform diverse baselines with remarkable consistency across tasks and resolutions. We hope that this strategy will prove useful for tasks requiring long rollouts and physical consistency, such as predicting atmospheric or ocean dynamics.

**Architectures and Hyperparameters** HREs proved effective with mUnet processors on rectangular grids, but our approach is orthogonal to $\mathcal{M}_{\text{HR}}$'s architecture so long as physics-respecting coarsening and interpolation operators are available. Which architectures benefit most from the HRE framework remains an open question, but a first step could be to consider our baselines and other successful emulators. Tuning of processor hyperparameters could prove beneficial, and the optimal values might not be the same as for the corresponding FRNEs. HREs could also be combined with hybrid solvers or numerical advection of $c_t$.

**Spatial Structure** Weather forecasting models operating on spherical mesh data [Lam et al., 2023, Oskarsson et al., 2024, Price et al., 2024] compute coarsened internal representations using graph-based message passing across resolutions, an approach compatible with physical and symmetry constraints [Horie and Mitsume, 2024]. However, mesh-based weather models have not yet included coarsened input fields, hard physical constraints, or multistep rollouts within a latent space. Similarly, cross-attention-based perceiver layers can drastically decrease token counts relative to the input dimensionality [Alkin et al., 2024]. These 'coarsening-like' operators for meshes and point clouds could be used to extend HREs beyond regular grids.

**Domains and Boundaries** Our PDEs were complex, but used simple spatial domains with periodic boundaries. Boundary conditions and geometric information could be provided as additional network inputs [Wandel et al., 2021, Lam et al., 2023, Watt-Meyer et al., 2025, Horie and Mitsume, 2024] to an HRE encoder. Such a general-purpose encoder could allow for an HRE foundation model.

**Generative Modeling** Our HREs and baselines use MSE losses. For long rollouts of chaotic dynamics, this can produce blurry outputs, distribution shift and instability. Generative models remedy this somewhat [Lippe et al., 2023, Kohl et al., 2024, Price et al., 2024, Zhou et al., 2024b], but reduce inference speed. Generative modeling with a loss like eq. 8 remains an open challenge.

**Further Constraints and Applications** HREs could also support hard symmetry constraints [Huang and Greenberg, 2025] or stability-inducing loss terms [Schiff et al., 2024]. Beyond our 2D fluid dynamics tasks, HREs could also be applied to 3D systems, higher field counts, solid mechanics, mixed-phase flows or spatially extended chemical and biological systems.

## Acknowledgments and Disclosure of Funding

We thank Nishant Kumar, Anthony Frion and Marcel Nonnenmacher for insightful comments on the manuscript. ACB, SA and CH were supported by the BMFTR grant *PLAGES* (16DKWN117B). ACB, SA and DSG were supported by Helmholtz AI and by the Helmholtz Foundation Model Initiative project *HClimRep*.

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

# A  Detailed model and training descriptions

## A.1  Additional training details

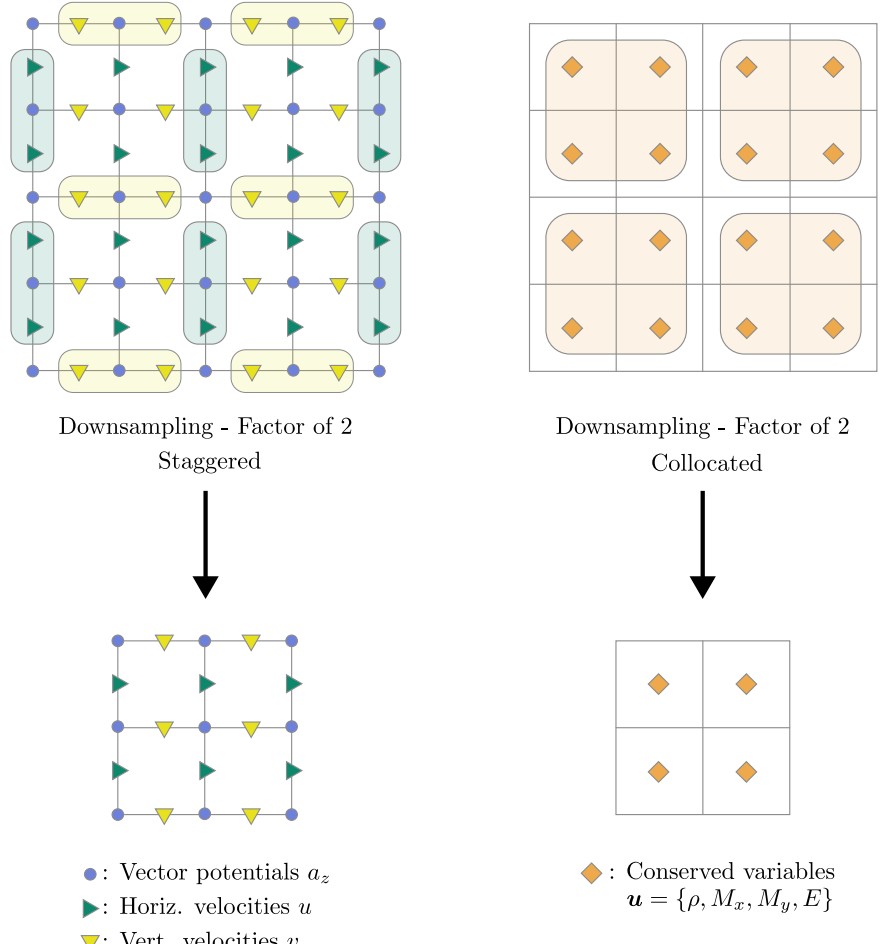

Figure 6: Coarsening operations for staggered and collocated grids. For staggered grids, we employ the face averaging approach for velocities described in [Kochkov et al., 2021]. This method conserves momentum and maintains incompressibility. In contrast, for collocated grids, we apply non-overlapping averaging that adheres to conservation laws.

We provide the summary of the training parameters in Table 2.

**Training of our decoder and Super-resolution (SR) models**  We train the SR models separately for upsampling tasks without unrolling in time. Since the rolled-out auxiliary variables $\tilde{c}_{t+k}$ are not expected to match the encoded variables $\mathcal{E}_\phi(X_{t+k})$, our decoder is trained on rolled-out trajectories of $(\tilde{x}_t, \tilde{c}_t)$. However, we stop gradient propagation to both the processor and encoder during decoder training, and thereby avoid propagating gradients backward in time. We note that there are no technical obstacles to training the HRE end-to-end in an encoder-processor-decoder framework as in eq. 4, we found this to be more costly and less effective in initial experiments.

## A.2  Additional model details

In this section, we give additional model details that are not present in the main text or Table 2. We also describe our modifications to the official implementations of FactFormer and DINo.

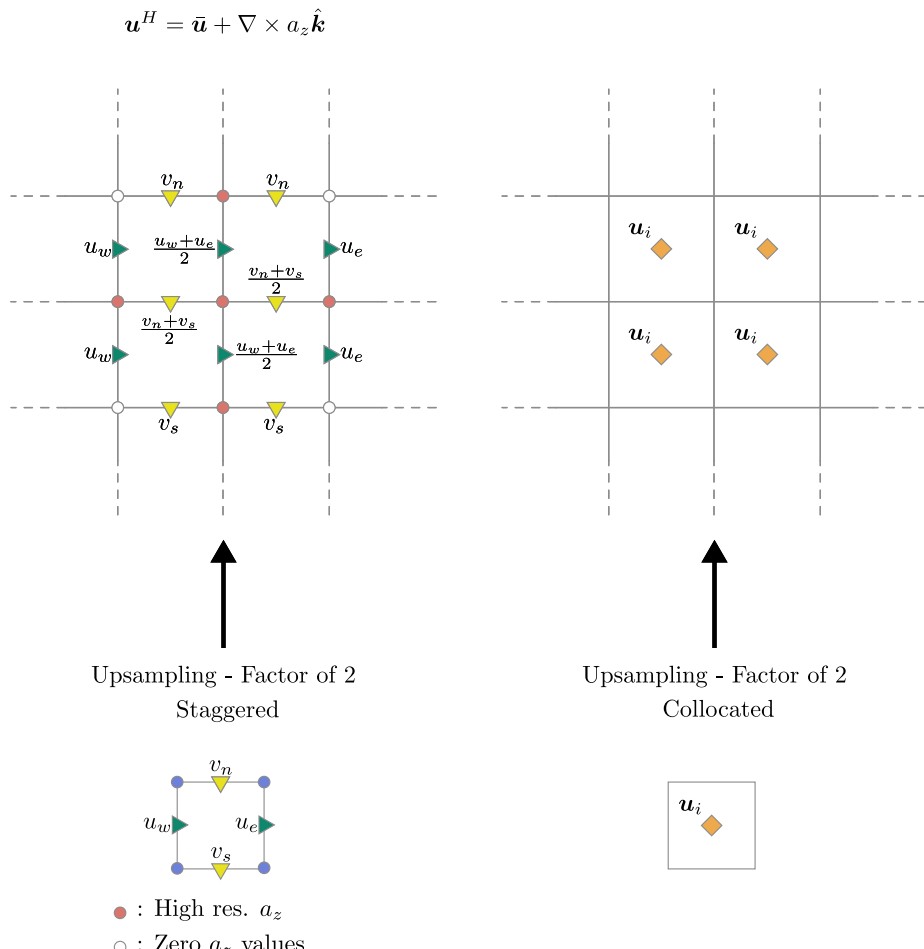

Figure 7: Upsampling methods for staggered variables and collocated grids. For staggered variables, face values are copied and velocity values in inner cells are linearly interpolated, preserving momentum conservation and incompressibility. Corrections are learned through a vector potential $a_z$. We either fix the corner values of $a_z$ to satisfy Equation 15 or relax this assumption to obtain continuous upsampled velocity fields. For collocated grids, upsampling is performed by copying pixels from the low-resolution input or by bilinearly interpolating them.

**mUnet**   Our implementation of mUnet uses an initial convolution layer to lift the channel dimension to 32. The 3 up/downsampling blocks contain 2 ConvNeXt blocks, linear attention [Shen et al., 2021] and a residual connection from the block inputs. The bottleneck layer includes multi-head self-attention [Vaswani et al., 2017], and skip connections join downsampling outputs to upsampling inputs at each resolution. At each level of the downsampling, two consecutive ConvNeXt layers [Woo et al., 2023] are followed by a 2D convolution using a stride 2 and a group norm with 32 groups. The channel dimensions for each level of downsampling are $[32, 64, 128, 256]$. Each skip connection incorporates linear self-attention operations. The bottleneck of the mUnet consists of a ConvNeXt layer followed by full self-attention and another ConvNeXt layer. The upsampling blocks are designed to be symmetric to the down blocks (We use transpose convolutions for upsampling).

**HRE ours**   Our implementation uses the up and down blocks of the mUnet without the skip connections. The encoder $\mathcal{E}_c$'s channel counts are $[32, 64, 128, 16]$ with corresponding norm groups of $[32, 32, 32, 8]$. The processor maintains the same hyperparameters as the mUnet but uses a different initial lifting convolution that operates on $(x_t, c_t)$ pairs. $c_t$ fields are passed through ConvNeXt layers before and after the processor, while $x_t$ fields are passed through a physical constraint layer after

the processor. Since in our high-resolution tasks the encoding and target resolutions are the same, the decoder is symmetric to the encoder except that the inputs are $(x_t, c_t)$ pairs and the first group normalization uses 9 groups for ID, KF and KFHR tasks and 10 groups for the CD task. Code is available at `https://github.com/alicanbekar/hres`.

**FactFormer**  We use the official FactFormer implementation, with a hidden dimension of 128, 4 layers, 8 heads, and kernel dimension 128. The input encoder is a 2D convolutional layer and the output decoder is an MLP. We increase the input channel count to 2. For rollout resolutions of $256^2$ and $512^2$, to prevent out-of-memory errors, we decrease the number of latent steps from 4 to 2, the output window from 16 to 10 and the batch size from 32 to 8. We also decrease the maximum rollout length for the pushforward trick in our training curriculum from 4 to 2. When evaluating accuracy for FactFormer-256 in Fig. 5, we compare to reference simulations coarsened to $256^2$. We note that despite this unfair advantage, FactFormer-256 remains less accurate than mUnet-64+HR and HRE-64.

**DINo**  We use a 3 layer decoder with 64 channels and a latent size of 100. ODE tendencies are computed with a 4-layer MLP with 512 hidden units. We adjust the number of input channels based on the emulation task. We observed that the performance of DINo after 12K epochs is not competitive, and instead trained it for 60K epochs with early stopping patience of 20K epochs.

**LatRoll-FAE and LatRoll-E2E**  Both models have the same setup as ours for rollout resolutions of $32^2$ and $64^2$. The only difference is that we remove the coarse $x_t$ fields and corresponding encoder-decoder pairs, and lift the latent fields to the same size before passing them to the propagator. During training and inference, these models take high-resolution inputs ($512^2$) and encode them into the latent space as $c_t = \mathcal{E}(X_t)$. The propagator advances the latent representation as $\tilde{c}_{t+k} = \mathcal{M}^k(c_t)$, and the decoder maps the propagated fields to a high-resolution vector potential $A^{t+k} = \mathcal{D}(\tilde{c}_{t+k})$. A curl layer at the output then calculates the high-resolution velocity fields $512^2$ as $X'_{t+k} = \nabla \times A^{t+k}$, satisfying the incompressibility condition. This approach maintains zero momentum in the output field, which is consistent with the initial conditions in the ID and KF cases where momentum is nearly zero. For the FAE model, we train the encoder-decoder pairs in autoencoder mode on the entire training set, then freeze these weights during unrolled training. All parameters of the E2E model are trained at each rollout step. We only minimize a loss between the network predictions and the ground truth at the final rollout step to keep the models trainable in a reasonable time. We use the same curriculum $K \in [1, 2, 4, 8, 16]$, with each curriculum step trained for 100 epochs. As a result, after the last curriculum step is completed, RMSE decreases for this model until 16 steps, then increases thereafter.

**DPOT-S**  This model uses Fourier attention on 1024-dimensional internal representation at rollout resolution $32^2$ or $64^2$, with 6 layers and 8 heads. We use the pretrained model and finetune it for the downstream ID and KF tasks using batch size 32 on a single H100 GPU. We train the model for 500 epochs on the same set of simulation trajectories used for the other baselines. In each epoch, each trajectory contributes a sample to one element of one batch, starting on a random time step of the trajectory. This results in training on a total of 960000 samples for KF and 640000 samples for ID in total, which is roughly equivalent to training for 35 epochs for ID and 20 epochs for KF on the whole dataset. We consider this a reasonable (or somewhat generous) quantity of data when finetuning a foundation model for few-shot inference. Attempts to use DPOT-S in zero-shot mode did not produce useful results beyond 5-10 time steps.

**DilResNet**  The architecture consists of a linear projection that maps PDE fields onto 48 channels, followed by 4 blocks, of 7 convolutional layers with $3 \times 3$ kernels. Each block uses kernel dilations of 1, 2, 4, 8, 4, 2 and 1. A final convolution maps onto output predictions. We use Swish activations, preceded by group normalization layers. For unrolled training, we use the same architecture on our standard curriculum $K \in [1, 2, 4, 8, 16]$. The batch size per GPU is fixed at 32, and the number of GPUs for each curriculum step is $(4, 6, 6, 6, 16)$. All other parameters match the DilResNet baseline with input noise training.

**FNO**  We use the official FNO implementation from [Kossaifi et al., 2024] (version 0.3.0), with 4 Fourier layers including lifting, filtering and projection, and hidden layers with 128 channels. We use 16 x- and y-modes for a parameter count close to the mUnet.

### A.3 Physics-constrained decoders for hybrid representations

Here we describe the construction of decoders with hard physical constraints. We consider two cases: conservation of multiple quantities on a collocated grid, and incompressibility with conservation of momentum on a staggered grid. In each case, we learn a correction $\mathcal{B}_c$ that is added to an interpolation operator $\mathcal{D}_x$. $\mathcal{D}_x$ is the coarse-to-fine counterpart of $\mathcal{E}_x$ (Fig. 7), and obeys all relevant physical constraints for each PDE.

#### A.3.1 Collocated grid

We enforce conservation of the field variable $u$ for KS as well as conservation of mass, energy and momentum for compressible flow on collocated grids by using an $r \times r$ local average as $\mathcal{E}_x$, and bilinear interpolation as $\mathcal{D}_x$. The network learns an additive correction to the updated fields $\mathcal{B}_\psi$, from which the local $r \times r$ average is removed to produce a conservative update $\mathcal{B}_c$. To remove the local average, this average is first computed using $\mathcal{E}_x$, then removed at high resolution using $\mathcal{D}_x$:

$$\mathcal{B}_c(\tilde{x}_t, \tilde{c}_t) = \mathcal{B}_\psi(\tilde{x}_t, \tilde{c}_t) - \mathcal{D}_x \circ \mathcal{E}_x \circ \mathcal{B}_\psi(\tilde{x}_t, \tilde{c}_t) \tag{14}$$

Here $\mathcal{D}_x$, a fixed interpolation operator respecting physical constraints (e.g., bilinear or nearest neighbor interpolation), is the coarse-to-fine counterpart of $\mathcal{E}_x$ (See Fig. (6, 7)-right). $\mathcal{B}_\psi$ is an unconstrained mapping from $(\tilde{x}_t, \tilde{c}_t)$ onto an additive correction for $\tilde{X}_t$. In eq. 14 the second term ensures that $\mathcal{B}_c$ corrects the physically constrained quantity (e.g. density, momentum or energy) in a way that respects physical constraints, and ensures the decoder satisfies the consistency condition:

$$\forall x_t, c_t : \qquad \mathcal{E}_x \circ \mathcal{D}_{\text{HR}}(x_t, c_t) = x_t \tag{15}$$

For example, if $\mathcal{E}_x$ averages a density field over nonoverlapping $r \times r$ pixel squares, and $\mathcal{D}_x$ copies each pixel $r \times r$ times, then $\mathcal{B}_c$ has 0 mean on each $r \times r$ square of outputs.

#### A.3.2 Staggered grid

Our KF, KFHR and ID tasks are based on incompressible flow on a staggered grid. In this case, $\mathcal{E}_x$ applies face-averaging (Fig. 6) as in Kochkov et al. [2021]. To satisfy the divergence-free condition, the decoder compute the curl of a learned high-resolution vector potential:

$$\mathcal{D}_{\text{HR}}(\tilde{x}_t, \tilde{c}_t) = \nabla \times (D_\psi(\tilde{x}_t, \tilde{c}_t) \cdot \delta_r(i, j)) \tag{16}$$

Here, $i$ and $j$ are pixel indices of the high-resolution output. Function $\delta_r(i, j)$ fixes the corners of each $r \times r$ square on the learned vector potential to zero (Fig. 7-left). This guarantees that the decoder satisfies the eq. 15 and is the counterpart of face-averaging $\mathcal{E}_x$ in Fig. 6-left. The function $\delta_r(i, j)$ is defined as:

$$\delta_r(i, j) = \begin{cases} 0 & \text{if } i \bmod r = 0 \text{ and } j \bmod r = 0, \\ 1 & \text{otherwise.} \end{cases} \tag{17}$$

$\mathcal{D}_x$ thus interpolates horizontal velocities horizontally and vertical velocities vertically (Fig. 7). Both $\mathcal{E}_x$ and $\mathcal{D}_x$ trivially conserve the incompressibility condition (consider the divergence of each cell in Fig. 7, upper left).

We observe that fixing the corner vector potential values introduces discontinuities that the correction term cannot rectify. Therefore, we relax the assumption of Equation 15 and allow the decoder to output continuous vector potential fields. This relaxation does not violate the conditions of incompressibility or momentum conservation. However, the encoding and decoding processes compromise the immutability of the coarse field variables, meaning that repeated decoding-encoding changes the coarsened fields $x_t$.

## B Measurement of inference speed

We analyze inference times across varying batch sizes for the ID and KF cases at rollout resolutions of $32^2$ and $64^2$ on an A100 GPU with 40GB memory. We exclude the data transfer and encoding costs for DINo and our model to obtain the marginal rollout cost per time step. To obtain the scaling behavior, we fit a linear regression using least squares to the largest three batch size measurements

for each model and calculate the resulting slope (However, we use the inference speed for batch size of 32 in Figure 1 due to GPU memory limitations). This slope approximates the per-batch-element inference time under GPU saturation conditions (considering that the last three points lie approximately on a line), while the y-intercept represents the computational overhead inherent to each model. The complete results are presented in Figures 23 and 24. In both cases, DilResNet is the slowest and DINo is the fastest model. Also, DINo has the smallest overhead while DilResnet has the highest. Our model is slightly slower than mUnet and slightly faster than FactFormer. All models are faster than the numerical solver operating in the native resolution, which takes $\approx 19.5$ seconds to solve the system. However, the numerical solver operating in $512^2$ input size takes $\approx 0.6$ seconds, meaning that models operating at $64^2$ rollout resolution with batch size 1 can be slower than this low-resolution numerical solver.

## C  Additional dataset details

In this section we give full details on the datasets used for each emulation task. All datasets are stored in `float32` precision. Additional information and dataset parameters are summarized in Table 4. Initial conditions are taken from [Kochkov et al., 2021] for ID, and from [Takamoto et al., 2022] for CD. For KF and KFHR, we take independent samples from the stationary distribution. Training, validation and testing data consisted of nonoverlapping sets of samples drawn from the same distributions (we use non-overlapping sets of random initialization keys for the training, validation, and testing datasets when generating data from JAX solvers), except in cases where we specifically measure generalization of trained emulators to out-of-distribution data (Table 9).

### C.1  2D Kolmogorov Flow (KF)

We use the staggered conservative DNS solver from the jax-cfd package [Kochkov et al., 2021] to integrate this system with periodic boundary conditions. We generate 128 experiments using $2048 \times 2048$ cells for the solver. We set the numerical solver's CFL [Courant et al., 1928] condition to 0.5 and then uniformly subsample its solution, obtaining $n_t = 300$ points in the interval $[40, 80]$. We then coarsen the data (velocities) to a $512 \times 512$ grid using the face-averaging approach to maintain incompressibility and momentum conservation. This $512 \times 512$ data then comprises the high-resolution inputs $\mathcal{A}_{r'}(X_t)$ for HREs and other methods operating on inputs at $512^2$ resolution for our emulation tasks, or are coarsened further as needed by other emulators. These spatial and temporal coarsening steps result in a CFL number of 9.4 for the training dataset, far exceeding that of the solver.

### C.2  2D Kolmogorov Flow with $\mathrm{Re} = 4000$ (KFHR)

The Reynolds number of KF scales as $\mathrm{Re} \propto \sqrt{\chi} L^{2/3}/\nu$. $\chi$ is the forcing strength scale, $L$ is the domain size and $\nu$ is the kinematic viscosity. Following [Kochkov et al., 2021] we scale $L \to 2L$, $\nu \to \nu/2$ and $\chi \to \chi/2$ to increase the Reynolds number to 4000. Since the smallest eddy size scales with $1/\sqrt{\mathrm{Re}}$, we get smaller eddy structures in the solution. We use a $4096 \times 4096$ grid for the solver and downsample the solution to a $512 \times 512$ grid. All other parameters match KF.

### C.3  2D Incompressible Decaying Turbulence (ID)

This dataset, like the Kolmogorov flow, is governed by the incompressible Navier-Stokes equations and solved with the same jax-cfd package. We generate 200 experiments using the same parameters and spatial resolutions as KF except for the forcing term, which is set to $\boldsymbol{f} = \boldsymbol{0}$. Time is uniformly discretized to $n_t = 166$ points in $[4.5, 25]$. The spectral density of the initial conditions is sampled from a log-normal distribution with variance 0.25 that peaks at a wavenumber of $u_{\max} = 4.2$. We then generate random initial conditions from a normal distribution and filter the fields using FFT to match the desired spectral density.

### C.4  Compressible Decaying Turbulence (CD)

We generate the dataset using direct numerical simulation (DNS) software provided by [Takamoto et al., 2022]. Time integration uses a mass, momentum and energy-conserving finite volume scheme,

whose implementation we slightly modified to avoid rounding errors when converting between floating point representations of momentum and velocity. Our dataset consists of 200 different simulations having varying initial conditions of density, momentum and energy. Space is uniformly discretized to $n_x = n_y = 512$ cells in $[0, 1] \times [0, 1]$ with periodic boundary conditions and time is uniformly discretized to $n_t = 100$ points in $[0, 5]$.

Initial velocities are generated as a superposition of sinusoidal waves, $\mathbf{v}(\mathbf{x}, t = 0) = \sum_{i=1}^{N} \mathbf{A} \sin(k\mathbf{x} + \phi_i)$ where $N = 4$, $\mathbf{A} = \bar{v}/|k|$, $\bar{v}$ is maximum initial velocity determined using the Mach number as $\bar{v} = c_s M$, where $c_s$ is the speed of sound. Wavenumbers are given by $k = 2\pi i/L$, where $L$ is domain length, and the phases $\phi_i \sim \mathcal{U}[0, 2\pi]$. Density and pressure are initialized uniformly.

### C.5  1D Kuramoto-Sivashinsky Equation

We generate the dataset using the jax-cfd package. The spatial domain (with length $L = 256$) is discretized into 2048 points, while the temporal domain $t \in [0, 150]$ is discretized into $n_t = 3000$ time steps. Initial conditions are specified as $u_0(x) = \sum_{m=1}^{5} A_m \sin(2\pi \ell_m x/L + \phi_m)$, where $\{A_m, \ell_m, \phi_m\}$ are random parameters sampled as in [Lippe et al., 2023]. Boundary conditions are specified as periodic. We use a spectral solver with an RK4 time integrator. To ensure statistical stationarity, we discard solutions up to $t = 80$ as a burn-in period. We generate 1000 trajectories and downsample the solutions by a factor of 5 along the temporal axis. Rollout resolution is chosen as $32^2$ meaning that $r = 64$.

## D  Supplementary experimental results

**KS and ID with multiple seeds**    To test the robustness of our HRE method to different initial seeds, we initialized the network used in the KS case on a coarse version of the dataset using 5 different random seeds and trained it on a next-step prediction task. We observed that the model's performance was consistent over seeds, with mean validation MSE of $5e - 8$ and standard deviation of $1e - 8$. We also repeated the ID experiment on a rollout resolution of $32^2$ with an additional seed, resulting in similar accuracy. The additional experiment is conducted with seed 44 and resulted in a validation rollout loss of 0.29. In comparison, the experiment we report was run with the seed 43 and achieved a rollout error of 0.31.

**TKE spectra**    We report the TKE spectra of PDE fields predicted by the trained models at a rollout resolution of $32^2$ in Figures 18-19. We observe that the high-resolution information encoded in $c_t$ fields enables our model to capture high-frequency features in the flow field. On the KF task, our model consistently outperforms others in reproducing the TKE spectrum of the reference solver. However, it occasionally produces unwanted oscillations at high frequencies for the ID case. For KF at rollout resolution $32^2$, we use the mUnet model trained on rollout length 16. The model trained on rollout length 2 overshoots some TKE fields by a large margin; we believe that training on longer rollouts helps the model in this case. However, for the stability plots on KF with rollout resolution $32^2$, we use the model trained on 2 steps since it achieves the highest correlation with the ground truth after 64 timesteps.

## E  Training Costs

For the ID case at rollout resolution $64^2$ and rollout length 16, mUnet requires $\approx 7$ hours to train versus $\approx 9.5$ hours for HRE on 8 A100s with batch size 16 per GPU. For long rollouts, most HRE computations occur at coarse resolution, but for short rollouts, mUnet is considerably faster: at rollout length 1, mUnet needs $\approx 1$ hour for 100 epochs on 4 A100s, while HRE requires $\approx 5$ hours on 8 A100s. FNO takes $\approx 20$ minutes to train on 4 A100s for rollout length 1 and $\approx 90$ minutes for rollout length 16. DINo trains in $\approx 7$ hours on a single V100 at the same input resolution with batch size 32. FactFormer takes $\approx 90$ minutes on a single H100 with batch size 32; however, it requires $\approx 106$ hours to train at rollout resolution $512^2$ with batch size 8. DilResNet takes $\approx 13$ hours to train for 500 epochs on 4 H100s in noise training mode with batch size 32. Finetuning for DPOT-S takes $\approx 1$ hour on an H100. LatRoll-FAE requires $\approx 9$ hours on 8 A100s for the KF case at rollout resolution 16 when training on a single timestep and $\approx 15$ hours on 16 timesteps, while LatRoll-E2E takes $\approx 15$

hours and ≈ 17 hours when training on rollouts of 1 and 16 timesteps, respectively. All batch sizes are per GPU.

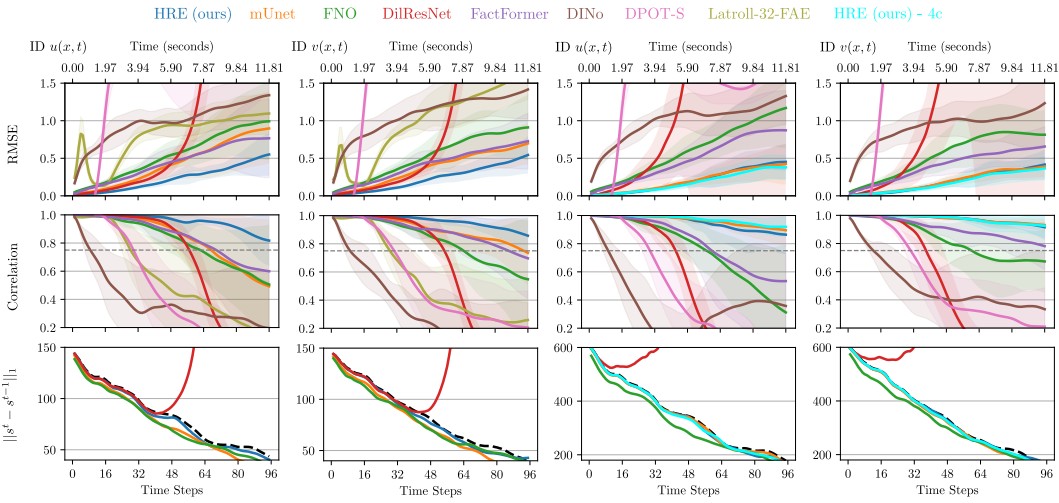

Figure 8: Rollout performance on the ID task at rollout/encoding resolutions $32^2$ (left), $64^2$.

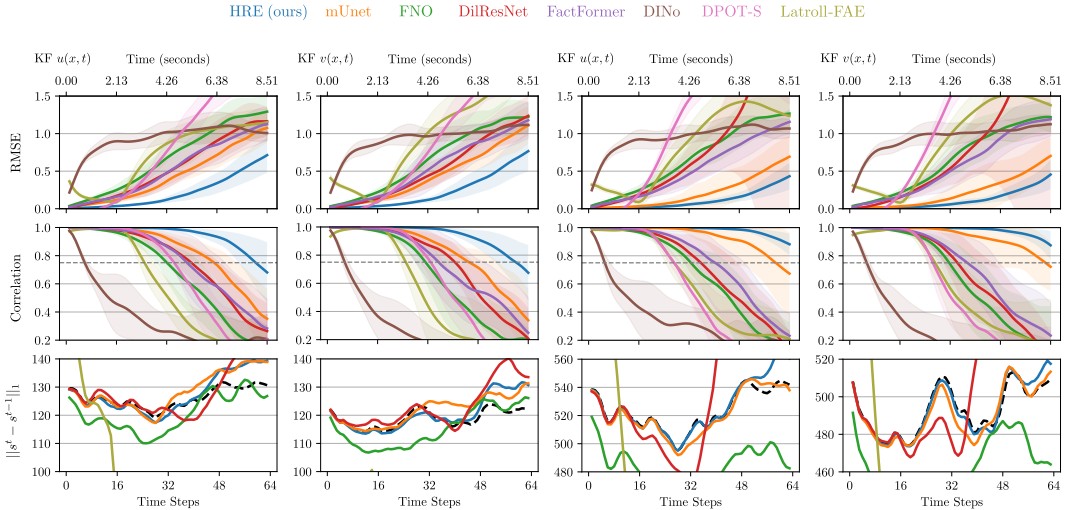

Figure 9: Rollout performance on the KF task at rollout/encoding resolutions $32^2$ (left) and $64^2$.

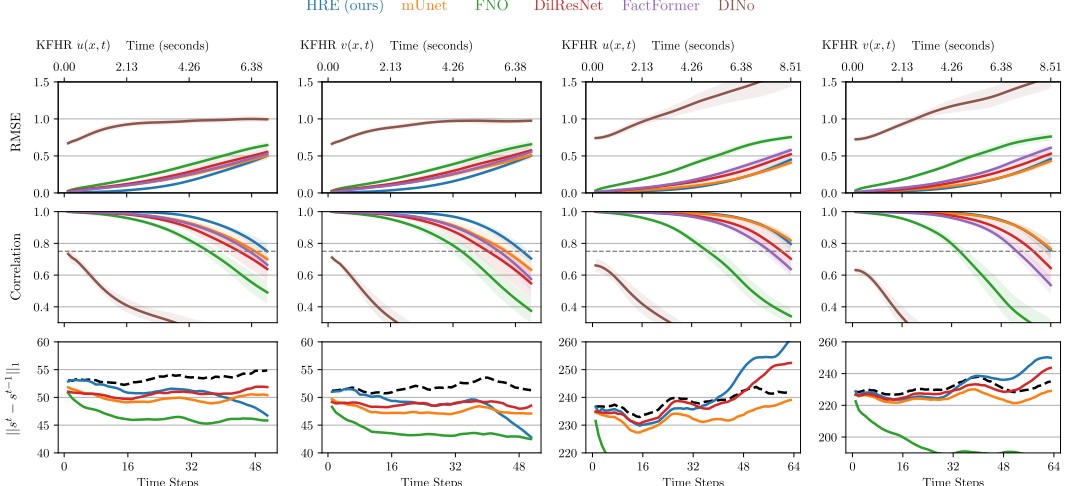

Figure 10: Rollout performance on the KFHR task at rollout/encoding resolutions $32^2$ (left) and $64^2$.

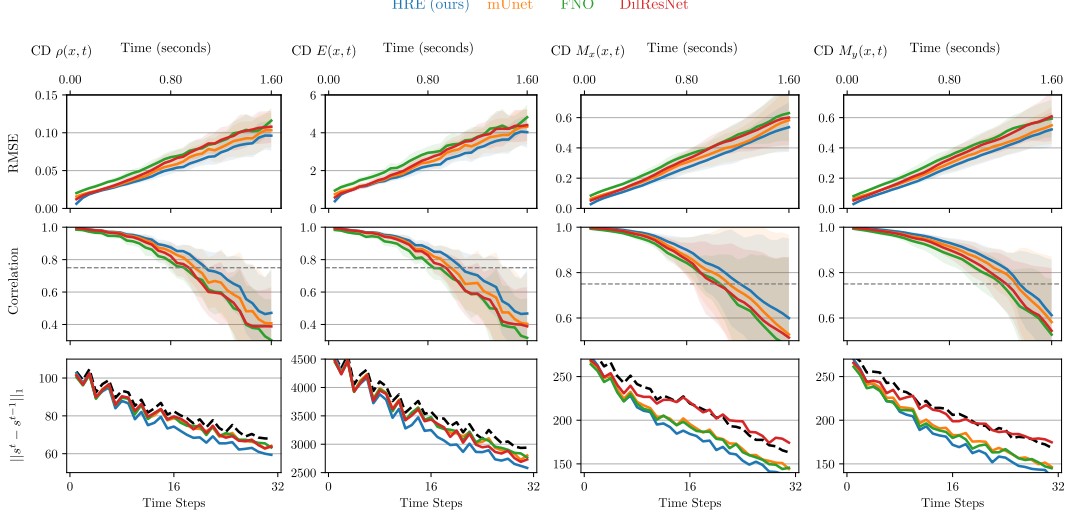

Figure 11: Rollout performance for $32^2$ CD.

Table 2: Training details. Our model trains with batch size 16 per GPU on KFHR; all other experiments use batch size 8. For mUnet, we maintain a fixed effective batch size across configurations by increasing the number of GPUs when the per-GPU batch size is decreased. FNO and mUnet models are trained on either A100s or H100s.

| | HRE (OURS) | mUNET | FNO | DILRESNET | FACTFORMER | DINO |
|---|---|---|---|---|---|---|
| | KF/KFHR/ID/CD | KF/KFHR/ID/CD | KF/KFHR/ID/CD | KF/KFHR/ID/CD | KF/ID/KFHR | KF/ID |
| BATCH SIZE (PER GPU) | 8/16 | 16/32 | 32 | 32 | 8/32 | 32 |
| INITIAL LEARNING RATE | $1e-4$ | $1e-4$ | $1e-4$ | $1e-4$ | $5e-4$ | $1e-2$ |
| OPTIMIZER | ADAMW | ADAMW | ADAMW | ADAMW | ADAMW | ADAM |
| WEIGHT DECAY | 0.01 | 0.01 | 0.01 | 0.01 | $1e-4$ | - |
| LR SCHEDULER | COSINE $\rightarrow 1e-6$ | COSINE $\rightarrow 1e-6$ | COSINE $\rightarrow 1e-6$ | COSINEWR $\rightarrow 1e-6$ | ONECYCLE | - |
| RESOURCES | 8 A100 GPUs | 8/4 A100/H100 GPUs | 4 A100/H100 GPUs | 4 H100 GPUs | 1 H100 GPU | 1 V100 GPU |
| CURRICULUM | $[1, 2, 4, 8, 16]$ | $[1, 2, 4, 8, 16]$ | $[1, 2, 4, 8, 16]$ | - | PUSHFORWARD $[1, 2, 3, 4]$ | ESS [16] |
| TRAINING NOISE | - | - | - | $\mu = 0, \sigma = 10^{-4}$ | - | - |
| TRAINING EPOCHS | $100 \times 5$ | $100 \times 5$ | $100 \times 5$ | 500 | 50 | 60000 |
| PADDING | CIRCULAR | CIRCULAR | CIRCULAR | CIRCULAR | - | - |
| LOSS | MSE | MSE | MSE | MSE | RELATIVE MSE | MSE |
| ACTIVATION | SiLU | SiLU | SiLU | SiLU | GELU | SiLU |
| EARLY STOPPING (75 EP.) | YES | YES | YES | YES | No | YES(20000 EP.) |
| SEED | 42 | 42 | 42 | 42 | 1234 | 1 |
| DEC./SR DEC. EPOCHS | 100 | 100 | - | - | - | - |

Table 3: Number of parameters per model.

|  | KF/ID | KFHR | CD |
|---|---|---|---|
| OUR PROCESSOR | 19.492M | 19.492M | 19.493M |
| mUNET | 19.430M | 19.430M | 19.433M |
| FNO | 18.974M | 18.974M | 18.974M |
| DILRESNET | 583680 | 583680 | 588096 |
| FACTFORMER | 3464834 | 3464834 | - |
| DINo | 769420 | 769420 | - |
| DPOT-S | 30M | - | - |
| LATROLL (FAE-E2E) $(64^2)$ | 23.8M | - | - |
| LATROLL (FAE-E2E) $(32^2)$ | 37.1M | - | - |
| OUR ENCODER $(64^2)$ | 2.156M | 2.156M | - |
| OUR ENCODER $(32^2)$ | 8.780M | 8.780M | - |
| OUR DECODER $(64^2)$ | 2.204M | 2.204M | - |
| OUR DECODER $(32^2)$ | 8.833M | 8.833M | - |
| SR DECODER $(64^2)$ | 2.153M | 2.153M | - |
| SR DECODER $(32^2)$ | 8.743M | 8.743M | - |

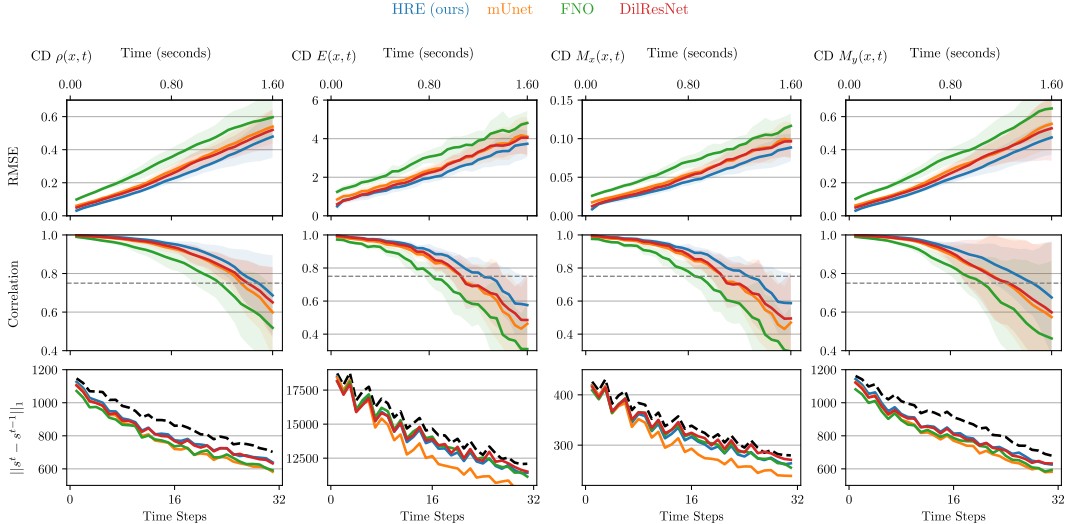

Figure 12: Rollout performance for $64^2$ CD.

Table 4: Dataset details.

| | | COMP. DECAYING | KOLMOGOROV | KOLMOGOROV (RE=4000) | INCOMP. DECAYING |
|---|---|---|---|---|---|
| | # SPATIAL DIMENSIONS | 2 | 2 | 2 | 2 |
| | # FEATURES | 4 | 2 | 2 | 2 |
| | FEATURES | $\rho, M_x, M_y, E$ | $v_x, v_y$ | $v_x, v_y$ | $v_x, v_y$ |
| | PARAMETERS | $(\zeta, \eta, M) = (10^{-8}, 10^{-8}, 0.4)$ | $(\rho, \mathrm{Re}, \nu) = (1, 10^3, 10^{-3})$ | $(\rho, \mathrm{Re}, \nu) = (1, 4 \times 10^3, 5 \times 10^{-4})$ | $Re_{\mathrm{init}} = 1000$ |
| | FORCING | $\mathbf{0}$ | $\sin(4y)\hat{\mathbf{e}}_1 - 0.1\mathbf{u}$ | $0.5\sin(4y)\hat{\mathbf{e}}_1 - 0.1\mathbf{u}$ | $\mathbf{0}$ |
| GRID | DOMAIN SIZE | $[0,1] \times [0,1]$ | $[0, 2\pi] \times [0, 2\pi]$ | $[0, 4\pi] \times [0, 4\pi]$ | $[0, 2\pi] \times [0, 2\pi]$ |
| | SOLVER GRID SIZE | $1/512$ | $2\pi/2048$ | $2\pi/4096$ | $2\pi/2048$ |
| | DATASET GRID SIZE | $1/512$ | $2\pi/512$ | $2\pi/512$ | $2\pi/512$ |
| | SPATIAL COARSENING | $8, 16$ | $1, 8, 16$ | $8, 16$ | $1, 8, 16$ |
| TEMP. INFO | SOLVER TIMESTEP | $\approx 1.1 \times 10^{-4}$ | $2.191 \times 10^{-4}$ | $2.191 \times 10^{-4}$ | $3.652 \times 10^{-4}$ |
| | EMULATOR TIMESTEP | $0.05$ | $0.133$ | $0.133$ | $0.123$ |
| | TEMPORAL COARSENING | $\approx 450\times$ | $\approx 600\times$ | $\approx 600\times$ | $\approx 340\times$ |
| | SOLVER CFL | $0.3$ | $0.5$ | $0.5$ | $0.5$ |
| | COARSE DATASET CFL | $28.125/56.25$ | $9.375/18.75$ | $4.687/9.375$ | $6.25/12.5$ |
| | BURN-IN TIME | $1$ | $40$ | $40$ | $4.2$ |
| | TERMINAL TIME | $5$ | $80$ | $80$ | $25$ |
| | TRAJECTORY LENGTH | $100$ | $300$ | $300$ | $166$ |
| PARTITIONS | # TRAJECTORIES TRAIN | $200$ | $128$ | $128$ | $200$ |
| | VALIDATION FRACTION | $0.1$ | $0.1$ | $0.1$ | $0.1$ |
| | # TRAJECTORIES TEST | $20$ | $16$ | $16$ | $16$ |

Table 5: Rollout performance across emulators and experiments. We report MSE and correlation (Corr.) of field variable $u$ for each method after 64 rollout steps (52 steps for KFHR at $32^2$). Best model performance in **bold** and second underlined.

| | KF | | ID | | KFHR | |
|---|---|---|---|---|---|---|
| | RMSE ↓ | CORR. ↑ | RMSE | CORR. | RMSE | CORR. |
| **ROLLOUT RESOLUTION** $32 \times 32$ | | | | | | |
| OURS | $\mathbf{0.71}_{\pm.23}$ | $\mathbf{0.68}_{\pm.21}$ | $\mathbf{0.29}_{\pm.08}$ | $\mathbf{0.96}_{\pm.03}$ | $\mathbf{0.50}_{\pm.04}$ | $\mathbf{0.74}_{\pm.05}$ |
| MUNET | $1.07_{\pm.20}$ | $\underline{0.35}_{\pm.21}$ | $\underline{0.58}_{\pm.34}$ | $0.78_{\pm.25}$ | $\underline{0.52}_{\pm.05}$ | $\underline{0.69}_{\pm.05}$ |
| FNO | $1.29_{\pm.21}$ | $0.12_{\pm.19}$ | $0.71_{\pm.27}$ | $0.74_{\pm.20}$ | $0.66_{\pm.05}$ | $0.48_{\pm.08}$ |
| DILRESNET | $1.16_{\pm.20}$ | $0.26_{\pm.20}$ | $1.77_{\pm.72}$ | $0.50_{\pm.17}$ | $0.55_{\pm.05}$ | $0.63_{\pm.07}$ |
| FACTFORMER | $1.12_{\pm.21}$ | $0.28_{\pm.20}$ | $\underline{0.58}_{\pm.25}$ | $\underline{0.79}_{\pm.51}$ | $0.53_{\pm.05}$ | $0.66_{\pm.05}$ |
| DINO | $\underline{1.00}_{\pm.10}$ | $0.20_{\pm.17}$ | $1.10_{\pm.27}$ | $0.30_{\pm.30}$ | $0.99_{\pm.04}$ | $0.21_{\pm.04}$ |
| DPOT-S | $1.85_{\pm.24}$ | $0.20_{\pm.17}$ | $1.65_{\pm.64}$ | $0.18_{\pm.24}$ | - | - |
| DILRESNET-UT | $1.19_{\pm.22}$ | $0.26_{\pm.22}$ | $0.84_{\pm.19}$ | $0.72_{\pm.16}$ | - | - |
| LATROLL-32-FAE | $1.15_{\pm.17}$ | $0.14_{\pm.16}$ | $0.96_{\pm.22}$ | $0.39_{\pm.35}$ | - | - |
| LATROLL-32-E2E | $1.23_{\pm.17}$ | $0.20_{\pm.16}$ | $1.20_{\pm.38}$ | $0.27_{\pm.43}$ | - | - |
| **ROLLOUT RESOLUTION** $64 \times 64$ | | | | | | |
| OURS | $\mathbf{0.43}_{\pm.17}$ | $\mathbf{0.88}_{\pm.07}$ | $\mathbf{0.28}_{\pm.15}$ | $\underline{0.93}_{\pm.12}$ | $\underline{0.44}_{\pm.05}$ | $\underline{0.79}_{\pm.04}$ |
| MUNET | $\underline{0.69}_{\pm.28}$ | $\underline{0.67}_{\pm.29}$ | $\mathbf{0.28}_{\pm.13}$ | $\mathbf{0.95}_{\pm.04}$ | $\mathbf{0.41}_{\pm.04}$ | $\mathbf{0.81}_{\pm.03}$ |
| FNO | $1.27_{\pm.22}$ | $0.14_{\pm.19}$ | $0.78_{\pm.26}$ | $0.67_{\pm.20}$ | $0.76_{\pm.05}$ | $0.35_{\pm.07}$ |
| DILRESNET | $7.05_{\pm13.22}$ | $0.13_{\pm.12}$ | $20.80_{\pm31.37}$ | $0.04_{\pm.10}$ | $0.52_{\pm0.08}$ | $0.70_{\pm.10}$ |
| FACTFORMER | $1.15_{\pm.15}$ | $0.23_{\pm.16}$ | $\underline{0.66}_{\pm.36}$ | $0.71_{\pm.37}$ | $0.58_{\pm.05}$ | $0.64_{\pm.05}$ |
| DINO | $1.07_{\pm.12}$ | $0.15_{\pm.12}$ | $1.09_{\pm.28}$ | $0.31_{\pm.33}$ | $1.67_{\pm.24}$ | $0.06_{\pm.10}$ |
| DPOT-S | $1.80_{\pm.40}$ | $0.09_{\pm.16}$ | $1.5_{\pm.27}$ | $0.17_{\pm.21}$ | - | - |
| DILRESNET-UT | $3.32_{\pm1.62}$ | $0.07_{\pm.17}$ | $6.56_{\pm3.58}$ | $0.05_{\pm.20}$ | - | - |
| LATROLL-64-FAE | $1.23_{\pm.23}$ | $0.21_{\pm.18}$ | - | - | - | - |
| LATROLL-64-E2E | $1.07_{\pm.16}$ | $0.32_{\pm.13}$ | - | - | - | - |

Table 6: Rollout performance across different experiments for the field variable $v$ for each method. Model parameters are the same as Table 5.

| METHOD | KF | | ID | | KFHR | |
|---|---|---|---|---|---|---|
| | RMSE↓ | CORR. ↑ | RMSE | CORR. | RMSE | CORR. |
| **ROLLOUT RESOLUTION $32 \times 32$** | | | | | | |
| OURS | $\mathbf{0.76}_{\pm.27}$ | $\mathbf{0.67}_{\pm.20}$ | $\mathbf{0.30}_{\pm.11}$ | $\mathbf{0.96}_{\pm.03}$ | $\mathbf{0.50}_{\pm.04}$ | $\mathbf{0.70}_{\pm.04}$ |
| MUNET | $\underline{1.10}_{\pm.24}$ | $\underline{0.34}_{\pm.22}$ | $\underline{0.51}_{\pm.21}$ | $\underline{0.86}_{\pm.12}$ | $\underline{0.53}_{\pm.05}$ | $\underline{0.62}_{\pm.07}$ |
| FNO | $1.22_{\pm.21}$ | $0.20_{\pm.20}$ | $0.73_{\pm.29}$ | $0.76_{\pm.18}$ | $0.68_{\pm.05}$ | $0.37_{\pm.08}$ |
| DILRESNET | $1.24_{\pm.26}$ | $0.21_{\pm.22}$ | $2.18_{\pm.78}$ | $0.40_{\pm.16}$ | $0.57_{\pm.06}$ | $0.55_{\pm.10}$ |
| FACTFORMER | $1.17_{\pm.23}$ | $0.25_{\pm.25}$ | $0.54_{\pm.23}$ | $0.85_{\pm.13}$ | $0.55_{\pm.05}$ | $0.58_{\pm.08}$ |
| DINO | $1.11_{\pm.10}$ | $0.00_{\pm.12}$ | $1.27_{\pm.31}$ | $0.05_{\pm.36}$ | $0.98_{\pm.02}$ | $0.06_{\pm.08}$ |
| DPOT-S | $2.11_{\pm.40}$ | $0.15_{\pm.25}$ | $9.23_{\pm2.19}$ | $0.31_{\pm.27}$ | - | - |
| DILRESNET-UT | $1.22_{\pm.19}$ | $0.20_{\pm.17}$ | $0.77_{\pm.19}$ | $0.80_{\pm.10}$ | - | - |
| LATROLL-32-FAE | $2.10_{\pm.31}$ | $0.10_{\pm.18}$ | $1.25_{\pm.32}$ | $0.34_{\pm.31}$ | - | - |
| LATROLL-32-E2E | $1.29_{\pm.24}$ | $0.13_{\pm.31}$ | $1.24_{\pm.36}$ | $0.43_{\pm.32}$ | - | - |
| **ROLLOUT RESOLUTION $64 \times 64$** | | | | | | |
| OURS | $\mathbf{0.45}_{\pm.19}$ | $\mathbf{0.87}_{\pm.10}$ | $\mathbf{0.27}_{\pm.14}$ | $\mathbf{0.97}_{\pm.03}$ | $\underline{0.46}_{\pm.06}$ | $\underline{0.76}_{\pm.05}$ |
| MUNET | $\underline{0.70}_{\pm.25}$ | $\underline{0.72}_{\pm.17}$ | $\underline{0.29}_{\pm.10}$ | $\mathbf{0.97}_{\pm.02}$ | $\mathbf{0.43}_{\pm.04}$ | $\mathbf{0.77}_{\pm.05}$ |
| FNO | $1.22_{\pm.20}$ | $0.18_{\pm.26}$ | $0.79_{\pm.38}$ | $0.72_{\pm.23}$ | $0.79_{\pm.05}$ | $0.21_{\pm.10}$ |
| DILRESNET | $5.02_{\pm8.14}$ | $0.11_{\pm.15}$ | $12.23_{\pm13.91}$ | $0.08_{\pm.12}$ | $0.53_{\pm.07}$ | $0.64_{\pm.08}$ |
| FACTFORMER | $1.19_{\pm.21}$ | $0.23_{\pm.23}$ | $0.54_{\pm.20}$ | $\underline{0.87}_{\pm.08}$ | $0.61_{\pm.04}$ | $0.54_{\pm.06}$ |
| DINO | $1.12_{\pm.11}$ | $0.03_{\pm.16}$ | $0.99_{\pm.27}$ | $0.42_{\pm.30}$ | $1.40_{\pm.14}$ | $0.04_{\pm.05}$ |
| DPOT-S | $3.63_{\pm.53}$ | $0.03_{\pm.26}$ | $9.42_{\pm1.49}$ | $0.32_{\pm.22}$ | - | - |
| DILRESNET-UT | $2.60_{\pm.96}$ | $0.12_{\pm.14}$ | $7.40_{\pm1.80}$ | $0.08_{\pm.11}$ | - | - |
| LATROLL-64-FAE | $1.38_{\pm.36}$ | $0.17_{\pm.32}$ | - | - | - | - |
| LATROLL-64-E2E | $1.02_{\pm.21}$ | $0.33_{\pm.22}$ | - | - | - | - |

Table 7: Rollout performance across different numbers of learned latent fields $n$ and coarsening factors $r$. We report MSE and correlation (Corr.) of field variable $u$ for each method. The rollout length is 96 steps for KF and 128 for ID. Values in **bold** show the best result at each rollout resolution.

| | KF | | ID | |
|---|---|---|---|---|
| | RMSE↓ | CORR. ↑ | RMSE | CORR. |
| **ROLLOUT RESOLUTION $32 \times 32$** | | | | |
| $n = 4$ | $\mathbf{1.263} \pm 0.23$ | $0.143 \pm 0.157$ | $0.6 \pm 0.319$ | $\mathbf{0.766} \pm 0.332$ |
| $n = 16$ | $1.295 \pm 0.188$ | $\mathbf{0.147} \pm 0.118$ | $\mathbf{0.192} \pm 0.227$ | $0.658 \pm 0.33$ |
| **ROLLOUT RESOLUTION $64 \times 64$** | | | | |
| $n = 4$ | $1.23 \pm 0.21$ | $0.214 \pm 0.215$ | $1.17 \pm 0.26$ | $0.868 \pm 0.192$ |
| $n = 16$ | $\mathbf{0.69} \pm 0.25$ | $\mathbf{0.69} \pm 0.24$ | $\mathbf{0.28} \pm 0.126$ | $\mathbf{0.95} \pm 0.039$ |

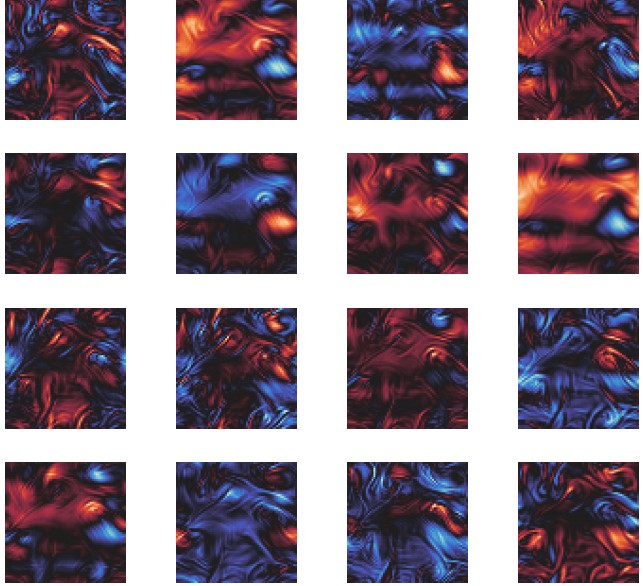

Figure 13: A snapshot of extracted $c_t$ fields for KF.

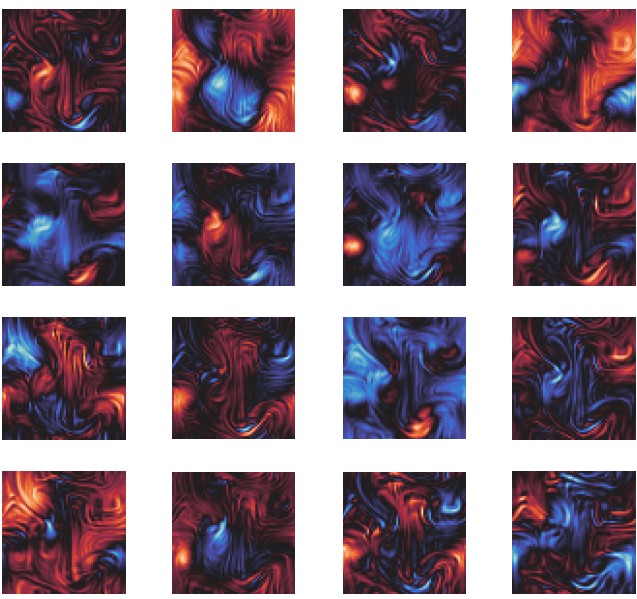

Figure 14: A snapshot of extracted $c_t$ fields for ID.

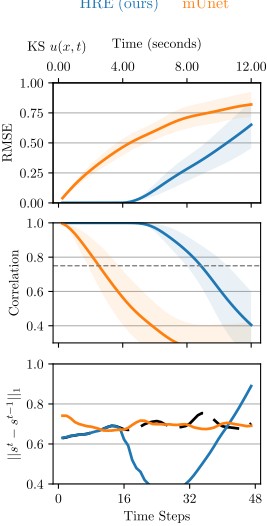

Figure 15: Comparison of mUnet emulators and HREs on the KS task. The reference resolution was 2048, and we used $r = 64$ for a rollout resolution of 32. We used $r' = 1$, meaning our HREs encoded full-resolution PDE fields from the reference simulation.

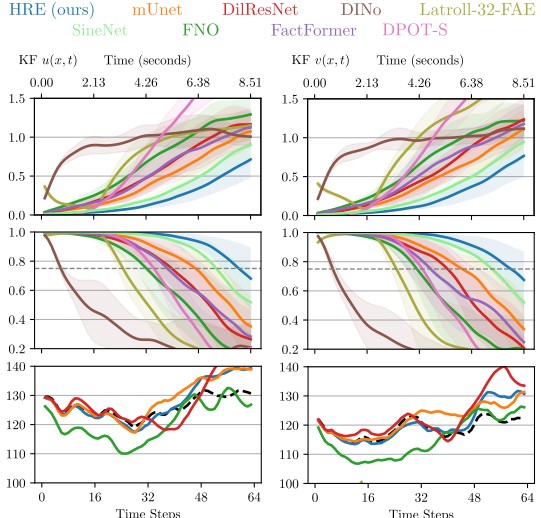

Figure 16: Comparison of SineNet, mUnet and HRE emulators on KF ($32^2$). We use the SineNet-8 model with the same training parameters as for their INS task [Zhang et al., 2024]. We excluded this baseline from further analysis due to its unfavorable cost-accuracy trade-off. While it provides a slight improvement in accuracy, it is several orders of magnitude slower than mUnet.

Table 8: Physics and $c_t$ field ablations. We measure the rollout performance of the models on KF ($32^2$) for 64 timesteps and compare the final RMSE and correlation values.

| | KF | |
|---|---|---|
| | RMSE | CORR. |
| OUR MODEL | $\mathbf{0.71} \pm 0.23$ | $\mathbf{0.68} \pm 0.21$ |
| MUNET W PHYSICS | $0.989 \pm 0.208$ | $0.455 \pm 0.213$ |
| OURS W/O PHYSICS | $0.881 \pm 0.21$ | $0.563 \pm 0.215$ |
| MUNET | $1.085 \pm 0.22$ | $0.345 \pm 0.21$ |

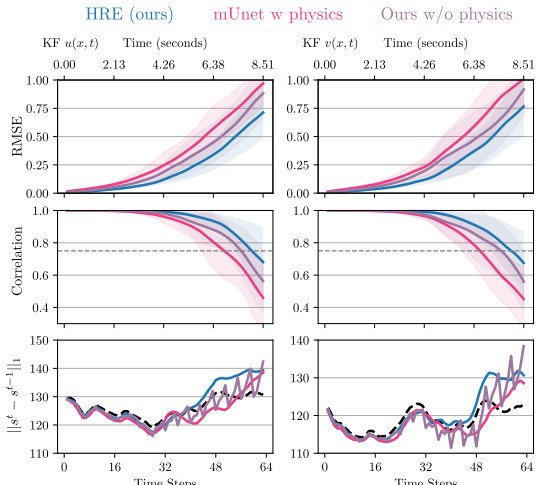

Figure 17: Performance of ablated models on KF $32^2$.

Table 9: Rollout performance of HREs vs. mUnet for varying IC distributions. The in-distribution training data were generated with $u_{\max} = 4.2$, and we vary this initial peak wavenumber to evaluate the generalization performance of our model versus mUnet. The burn-in time is calculated by scaling the default value for $u_{\max} = 4.2$ by the inverse ratio of the chosen peak wavenumber; all other parameters remain the same. Rollout and target resolutions are $32^2$. We report RMSE and correlation (Corr.) of the field variable $u$ after 64 rollout steps across increasing $u_{\max}$ values. Best performance in **bold**.

| | $u_{\max} = 3$ | | $u_{\max} = 4.2$ | | $u_{\max} = 5$ | | $u_{\max} = 6$ | |
|---|---|---|---|---|---|---|---|---|
| METHOD | RMSE $\downarrow$ | CORR. $\uparrow$ | RMSE | CORR. | RMSE | CORR. | RMSE | CORR. |
| HRE (OURS) | $\mathbf{0.15}_{\pm.05}$ | $\mathbf{0.98}_{\pm.02}$ | $\mathbf{0.29}_{\pm.08}$ | $\mathbf{0.96}_{\pm.01}$ | $\mathbf{0.65}_{\pm.31}$ | $\mathbf{0.85}_{\pm.13}$ | $\mathbf{1.01}_{\pm.55}$ | $\mathbf{0.71}_{\pm.34}$ |
| MUNET | $0.30_{\pm.12}$ | $0.90_{\pm.06}$ | $0.58_{\pm.34}$ | $0.78_{\pm.25}$ | $0.74_{\pm.42}$ | $0.75_{\pm.35}$ | $1.05_{\pm.68}$ | $0.61_{\pm.50}$ |

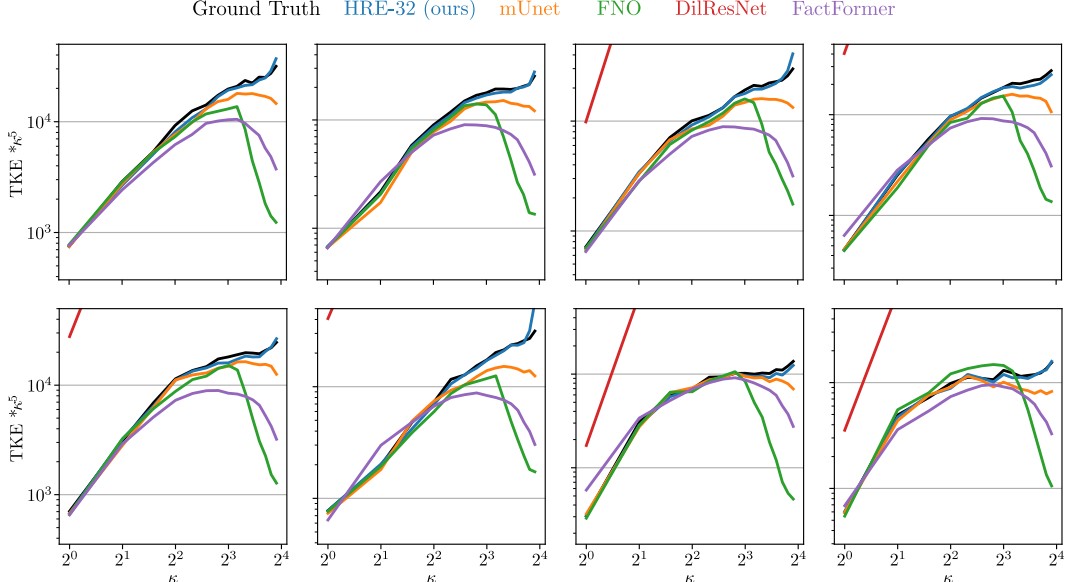

Figure 18: Additional TKE analysis for ID rollouts of 96 (Resolution $32^2$).

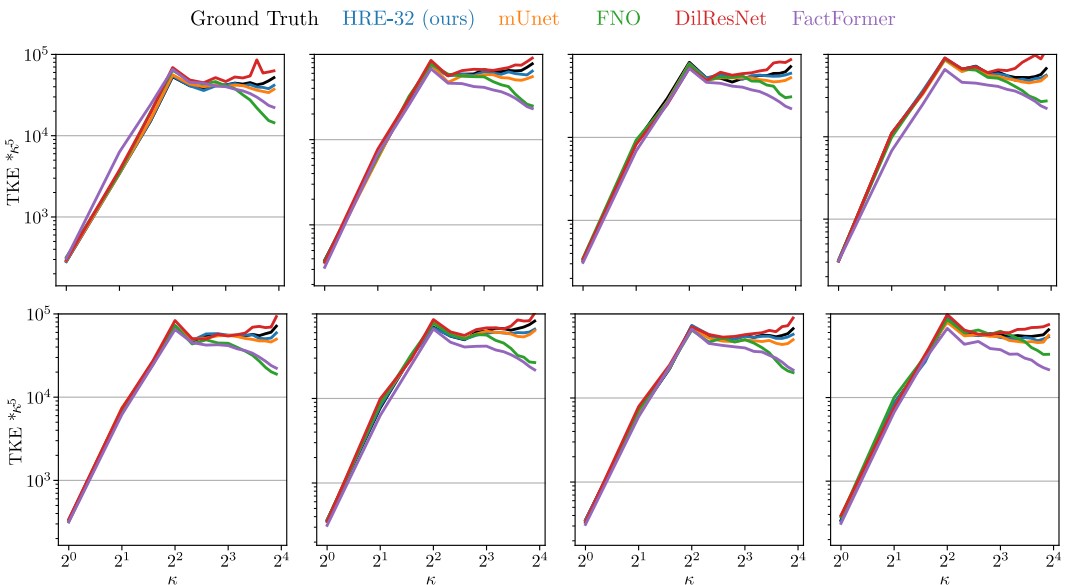

Figure 19: Additional TKE analysis for KF rollouts of 64 (Resolution $32^2$).

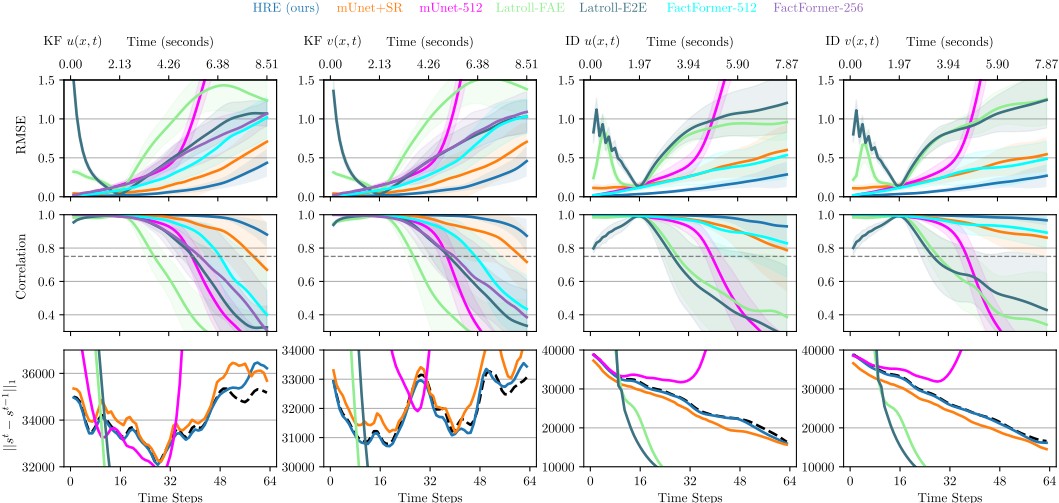

Figure 20: Comparison of our HRE with mUnet and FactFormer operating on high-resolution data and mUnet with a super-resolution decoder for the KF (left) and ID (right) tasks. Rollout resolutions are $64^2$ for KF and $32^2$ for ID for our model, mUnet+SR and Latroll models.

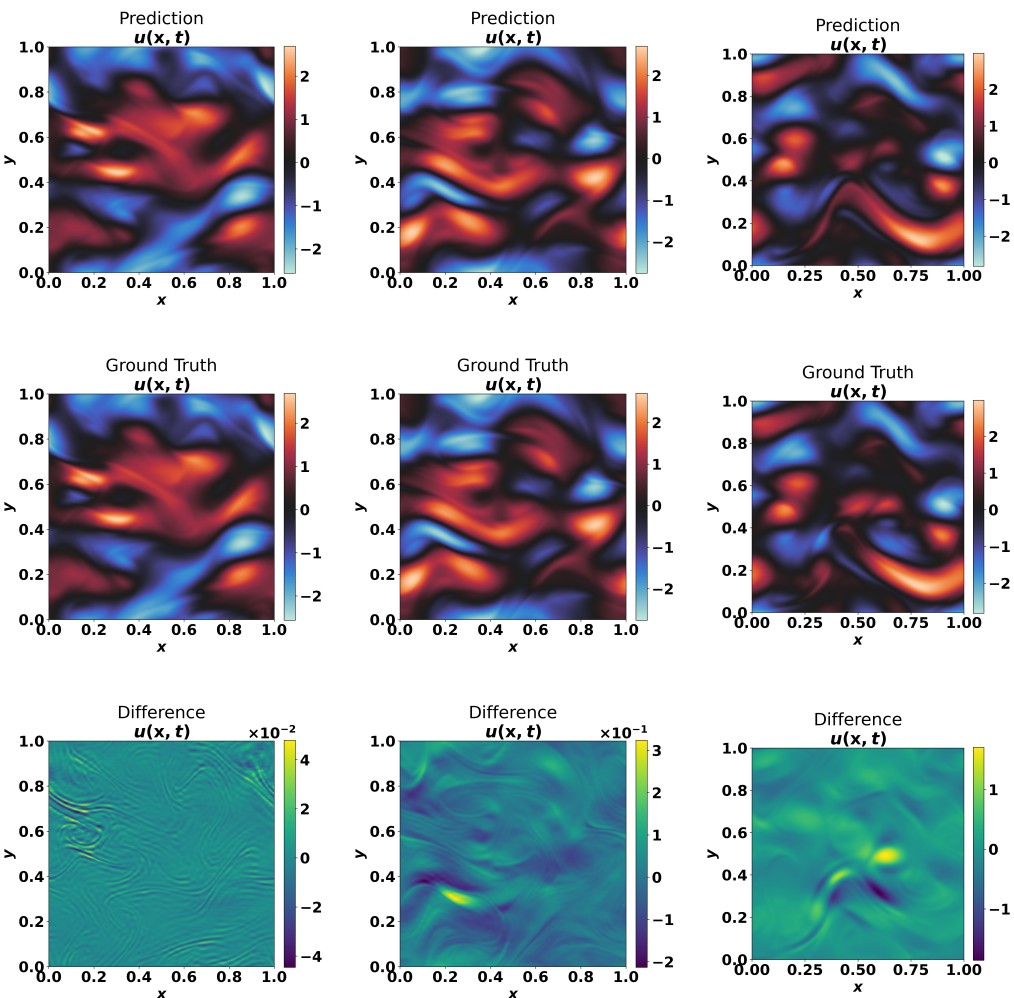

Figure 21: HRE-based prediction of high-resolution KF fields with rollout resolution $64^2$ and target resolution $512^2$ with 32 timestep intervals.

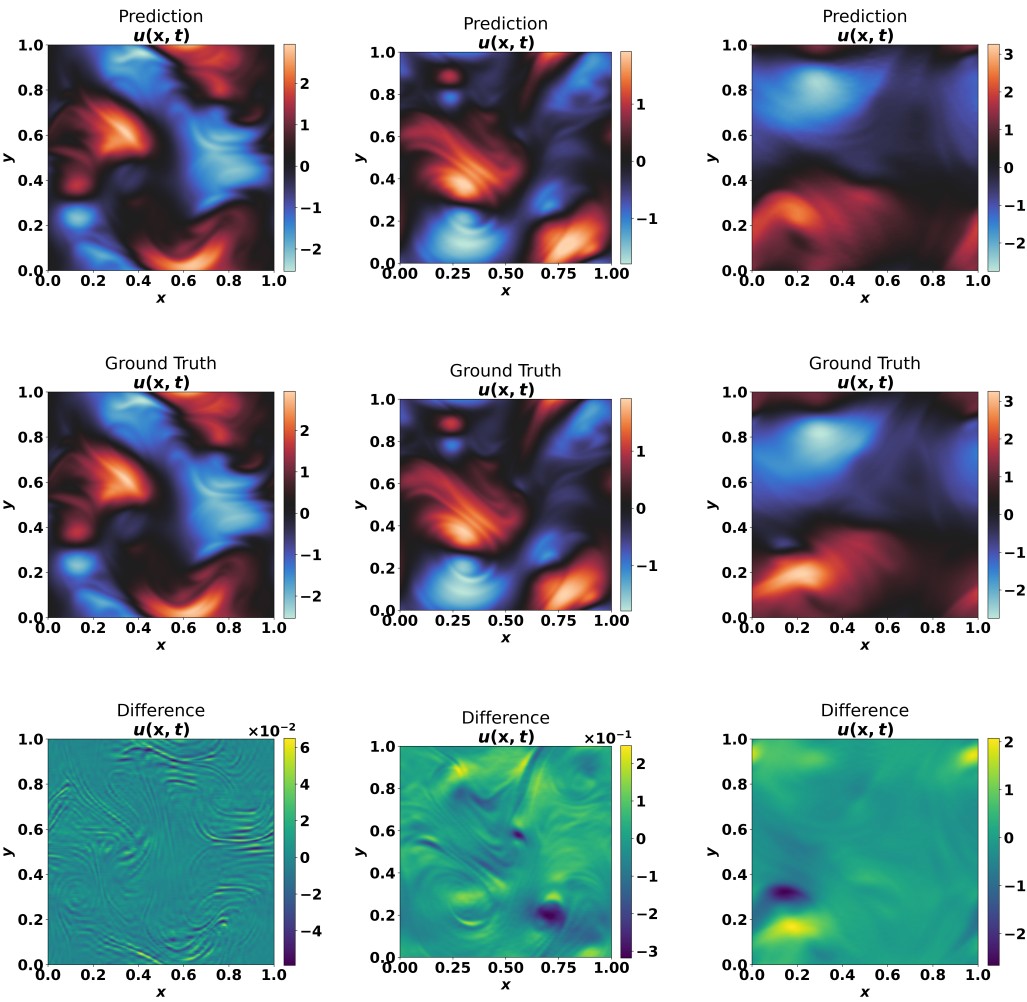

Figure 22: HRE-based prediction of high-resolution ID fields with rollout resolution $64^2$ and target resolution $512^2$ with 32 timestep intervals.

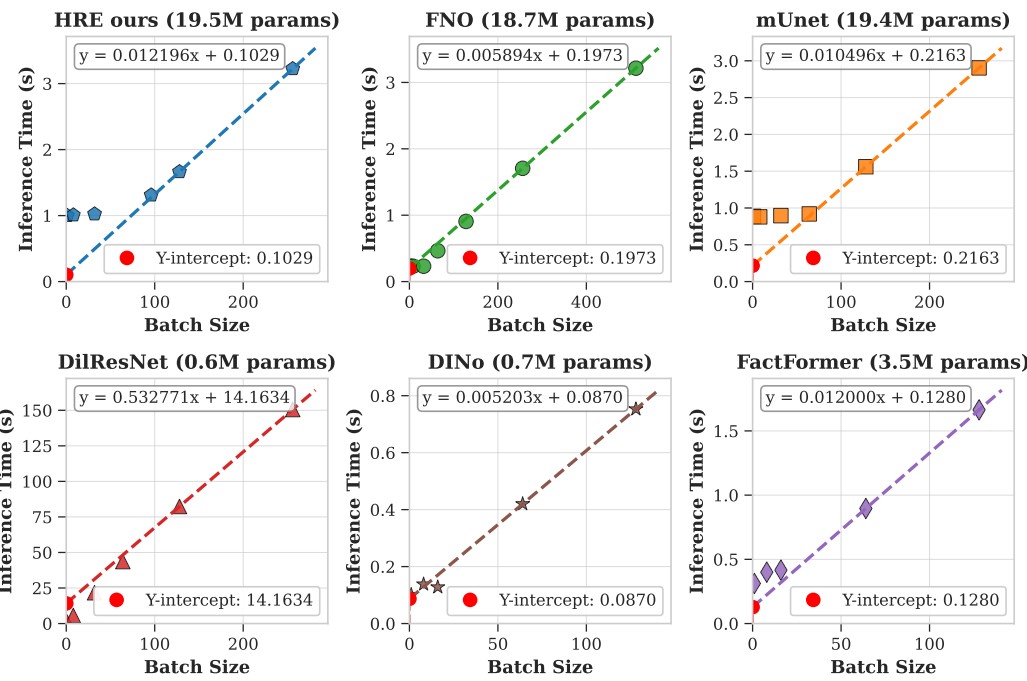

Figure 23: Inference times for varying batch sizes. We choose rollout resolution $(32^2)$ for ID, KF and KFHR tasks and computed rollouts over 64 time steps.

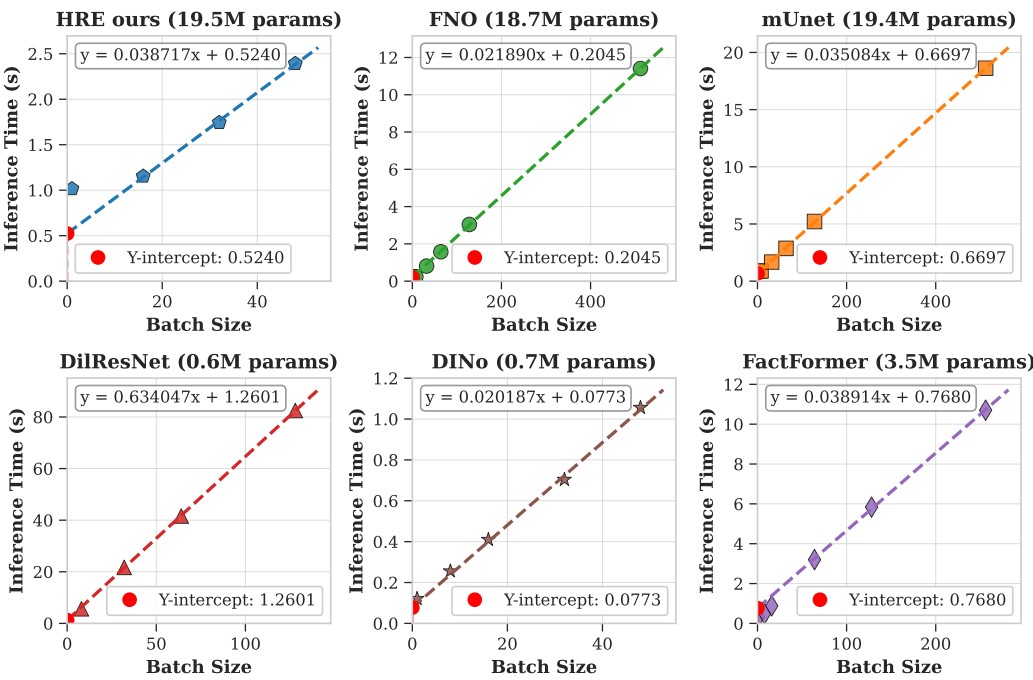

Figure 24: Inference times for varying batch sizes. We choose rollout resolution $(64^2)$ for ID, KF and KFHR tasks and computed rollout for 64 time steps.

| Benchmark | KF | | | | | | | | ID | | | | | | | KFHR | | | | CD | | KS |
|---|---|---|---|---|---|---|---|---|---|---|---|---|---|---|---|---|---|---|---|---|---|---|
| Rollout/Enc. Res. | 100 | | $32^2$ | | $64^2$ | | $256^2$ | $512^2$ | 100 | | $32^2$ | | $64^2$ | | $512^2$ | 100 | | $32^2$ | $64^2$ | $32^2$ | $64^2$ | $32^2$ |
| Target Res. | $32^2$ | $64^2$ | $32^2$ | $512^2$ | $64^2$ | $512^2$ | $256^2$ | $512^2$ | $32^2$ | $64^2$ | $32^2$ | $512^2$ | $64^2$ | $512^2$ | $512^2$ | $32^2$ | $64^2$ | $32^2$ | $64^2$ | $32^2$ | $64^2$ | $32^2$ |
| HRE (Ours) | | | $\underline{\checkmark}$ | | $\underline{\checkmark}$ | ✓ | | | | | $\underline{\checkmark}$ | ✓ | $\underline{\checkmark}$ | | | | | ✓ | ✓ | ✓ | ✓ | ✓ |
| mUnet | | | $\underline{\checkmark}$ | | $\underline{\checkmark}$ | ✓ | | $\underline{\checkmark}$ | | | $\underline{\checkmark}$ | ✓ | $\underline{\checkmark}$ | | $\underline{\checkmark}$ | | | ✓ | ✓ | ✓ | ✓ | ✓ |
| FNO | | | $\underline{\checkmark}$ | | $\underline{\checkmark}$ | | | | | | $\underline{\checkmark}$ | | $\underline{\checkmark}$ | | | | | ✓ | ✓ | ✓ | ✓ | |
| DilResNet-NT | | | $\underline{\checkmark}$ | | $\underline{\checkmark}$ | | | | | | $\underline{\checkmark}$ | | $\underline{\checkmark}$ | | | | | ✓ | ✓ | ✓ | ✓ | |
| DilResNet-UT | | | $\underline{\checkmark}$ | | $\underline{\checkmark}$ | | | | | | $\underline{\checkmark}$ | | $\underline{\checkmark}$ | | | | | | | | | |
| FactFormer | | | $\underline{\checkmark}$ | | $\underline{\checkmark}$ | | $\underline{\checkmark}$ | $\underline{\checkmark}$ | | | $\underline{\checkmark}$ | | $\underline{\checkmark}$ | | $\underline{\checkmark}$ | | | ✓ | ✓ | | | |
| DINo | $\underline{\checkmark}$ | $\underline{\checkmark}$ | | | | | | | $\underline{\checkmark}$ | $\underline{\checkmark}$ | | | | | | ✓ | ✓ | | | | | |
| DPOT-S | | | ✓ | | ✓ | | | | | | ✓ | | ✓ | | | | | | | | | |
| LatRoll-FAE | | | $\underline{\checkmark}$ | ✓ | $\underline{\checkmark}$ | ✓ | | | | | $\underline{\checkmark}$ | ✓ | | | | | | | | | | |
| LatRoll-E2E | | | $\underline{\checkmark}$ | ✓ | $\underline{\checkmark}$ | ✓ | | | | | $\underline{\checkmark}$ | ✓ | | | | | | | | | | |

Table 10: Summary of the trained models. We train models for specific rollout resolutions (encoding resolution for DINo and FactFormer) and target resolutions. When the target resolution exceeds the rollout resolution, we train a separate super-resolution decoder, indicated by "SR" in model suffixes (except for HRE models, which have their own decoders). When the target resolution is smaller than the rollout resolution, we coarsen the outputs to the target resolution using conservative coarsening operators. For clarity, models in the main text are named using their rollout resolution, e.g., mUnet-32+SR indicates that the model rolls out at a resolution of $32^2$. Underlined cases are used for the cost-accuracy scatter plot in Fig. 1 d . We use a rollout length of 96 for ID and 64 for KF cases on in-distribution held-out testing data.

