# OpenReview forum: "Hybrid Latent Representations for PDE Emulation"
_NeurIPS.cc/2025/Conference — NeurIPS 2025 poster_

### Official Review · Reviewer_6mAC · 2025-06-13

**Clarity:** 2
**Significance:** 2
**Originality:** 3
**Rating:** 4
**Confidence:** 4

**Summary:**

The authors present a method for training/rolling out neural emulators that combines the downsampled input field and an encoded representation of the original field. Experiments on coarse grids show that the additional context from high-resolution inputs aids in rollout performance, and operating in the low-resolution regime allows for fast inference.

**Questions:**

- As stated earlier, I would be happy to improve my rating if the authors can show that their method is better than using an equivalent, but purely latent approach.
- For Figure 5 (experiments on high-res data), how is the error calculated for Factformer-256 if the training data is 512x512?

**Ethical Concerns:**

["NO or VERY MINOR ethics concerns only"]

**Final Justification:**

- The perspective taken about downsampling and the resolution of PDE inputs is one that is not typically considered (usually left as a detail in the dataset generation or prior PDE benchmark datasets).
- The performance gain of using both a latent variable and the coarsened input is good.
- The clarity of the work could be improved, as it is currently difficult as a reader to understand the main message or the comparisons presented in the figures.

**Limitations:**

Many engineering simulations are mesh-based, which wouldn’t really admit a good averaging operator or perhaps even need coarsening. Some emulators can work with a full-resolution field/mesh, which would theoretically be better than an HRE approach.

**Paper Formatting Concerns:**

Line 184, small typo "anywher"

**Quality:**

3

**Strengths And Weaknesses:**

Strengths
- The problem is framed very well.
- The proposed method is explained well and makes a lot of intuitive sense.
- The experiments are executed well.
- It is good that the authors bring attention to details that are usually overlooked in PDE studies, in particular, the downsampling of simulation data.

Weaknesses
- I would say the main weakness is a lack of comparison to a convincing ULSE baseline. In particular, the speed/resolution of the HRE method can be achieved by ULSEs, and their proposed limitations don’t seem to be justified. Physics-based losses can be applied after decoding and spatial/structured inductive biases don’t seem to be necessary for PDE learning, as attention-based emulators are nearly SOTA (Transolver, UPT, etc.). If the authors could run a test on comparing the HRE w/ an equivalent architecture where the downsampled field is not used, specifically, use the same encoder to produce c = E(X), use a processor that propagates c instead of both (c, x), and use a decoder that only takes in c to produce X’ = D(c), this would be beneficial. Increasing the latent channels n only improves HRE performance, and at 16 latent channels and 3 physical channels (assume 2D velocity + pressure) is the original downsampled field needed anymore?
- At the rollout resolutions that HREs work the best (32^2), the resolution may be too coarse for practitioners to really use it, as many details are lost that may be important. At more useful rollout resolutions (64^2 or 128^2), the benefits are more incremental.

---

> ### Author Rebuttal · Authors · 2025-07-31
>
> > I would say the main weakness is a lack of comparison to a convincing ULSE baseline. In particular, the speed/resolution of the HRE method can be achieved by ULSEs, and their proposed limitations don’t seem to be justified.
>
> We agree that ULSEs are the fastest methods, since their lack of spatial structure leads to very compact latent spaces and inexpensive rollouts. We now better emphasize this in section 2.2.
>
> However, we were surprised by the assertion that a convincing ULSE baseline has not been included. DINo and FactFormer are ULSEs, recently published with multiple strong results.
>
> We are unaware of ULSE methods likely to be more effective than these on our tasks. Transolver and UPT (see response to 7kAd) are promising and interesting methods, but have not shown strong results on closed NS systems simulated at high resolutions for more than 10 time steps (UPT used a pipe flow, in which errors can be "washed out" by the flow, while Transolver only predicted 10 steps of NS). Given our many baselines (now 9) and finite compute, we chose DINo and FactFormer, which already showed strong results on tasks similar to ours.
>
> > Physics-based losses can be applied after decoding and spatial/structured inductive biases don’t seem to be necessary for PDE learning, as attention-based emulators are nearly SOTA (Transolver, UPT, etc.).
>
> Physics-derived losses are usually proven less effective than hard-constraints [1].
>
> > If the authors could run a test on comparing the HRE w/ an equivalent architecture where the downsampled field is not used, specifically, use the same encoder to produce c = E(X), use a processor that propagates c instead of both (c, x), and use a decoder that only takes in c to produce X’ = D(c), this would be beneficial. Increasing the latent channels n only improves HRE performance, and at 16 latent channels and 3 physical channels (assume 2D velocity + pressure) is the original downsampled field needed anymore?
>
> We trained these c-field-only models as suggested. To match the inference speed with our HREs, we used the same latent space resolution and total channel counts, and similar parameter counts. We removed the physical constraints from the processor and placed it at the output layer of the network as recommended.
>
> During training and inference, this new model takes high-resolution inputs (512x512) and encodes them into the latent space as $c_t = \mathcal E(X_t)$. The propagator then advances the latent representation as $\tilde c_{t+k} = \mathcal M^k(c_t)$, and the decoder maps these propagated fields to a high-resolution vector potential $A^{t+k} = \mathcal D(\tilde c_{t+k})$. A final curl layer then calculates the high-resolution velocity fields (512x512) as $X^\prime_{t+k} = \nabla \times A^{t+k}$, satisfying the incompressibility condition. Conservation of momentum is also satisfied since for our tasks the mean momentum is always zero, and this is achieved (as a hard constraint) by using a periodic vector potential. We train by minimizing MSE on high resolution predictions.
>
> We use the same curriculum $K \in [1, 2, 4, 8, 16]$, with each curriculum step trained for 100 epochs. We implemented two training strategies:
> 1. Train an autoencoder on $X_t$, freeze it then train the propagator with our rollout curriculum (LatRoll-FAE)
> 2. Train the entire network end-to-end at each curriculum step (LatRoll-E2E).
>
> For short rollouts, the performance of these models is comparable to HREs. For longer rollouts, HREs increasingly outperform them.
>
> We trained LatRoll-FAE and LatRoll-E2E on two datasets (ID and KF) with rollout resolution 32×32. The following tables show the RMSE and correlation values for rollout timesteps 16, 32, and 64 for our HREs versus these models.
>
> **Rollout performance of Ours vs Latent Rollout Models without coarse fields for KF dataset field variable u**
> | KF | Tsteps=16|  | Tsteps=32 |  | Tsteps=64 |  |
> |--|----|----|----|----|----|----|
> |  | RMSE ↓ | Corr. ↑ | RMSE ↓ | Corr. ↑ | RMSE ↓ | Corr. ↑ |
> | Ours | **0.03**±.01 | **0.99**±.00 | **0.13**±.04 | **0.99**±.01 | **0.71**±.10 | **0.69**±.27 |
> | LatRoll-E2E | 0.09±.02 | 0.99±.01 | 0.80±.16 | 0.66±.08 | 1.23±.18 | 0.20±.17 |
> | LatRoll-FAE | 0.09±.02 | 0.99±.00 | 0.87±.18 | 0.52±.11 | 1.15±.18 | 0.14±.17 |
>
>
> **Rollout performance of Ours vs Latent Rollout Models without coarse fields for ID dataset field variable u**
> | ID | Tsteps=16|  | Tsteps=32 |  | Tsteps=64 |  |
> |--|----|----|----|----|----|----|
> |  | RMSE ↓ | Corr. ↑ | RMSE ↓ | Corr. ↑ | RMSE ↓ | Corr. ↑ |
> | Ours | **0.05**±.01 | **0.99**±.00 | **0.12**±.03 | **0.99**±.00 | **0.30**±.09 | **0.96**±.03 |
> | LatRoll-E2E | 0.12±.03 | 0.99±.01 | 0.85±.15 | 0.63±.22 | 1.20±.39 | 0.27±.44 |
> | LatRoll-FAE | 0.13±.04 | 0.99±.01 | 0.78±.11 | 0.67±.22 | 0.96±.23 | 0.39±.36 |
>
> As mentioned (L133, L137, L157), we find training a latent space rollout model by directly minimizing MSE on high-resolution targets creates a conflict between long-term accuracy (our main aim) and high-resolution reconstruction. Our simple loss on $x_t$ (eq. 8) avoids this, prioritizing long-rollout accuracy as desired. These new results provide key empirical evidence to support these arguments.
>
> Additionally, training on MSE for high-resolution predictions is costlier than our HREs, since we use low-resolution targets.
>
> We thank the reviewer for raising this important question, and helping us to avoid overlooking an important ablation experiment. We now add this ablation experiment to the manuscript, including for the KFHR and CD cases. Overall, we find these new results significantly strengthen our manuscript and underline the importance of the hybrid nature of our learned representations for PDE system states.
>
> >At the rollout resolutions that HREs work the best (32^2), the resolution may be too coarse for practitioners to really use it, as many details are lost that may be important. At more useful rollout resolutions (64^2 or 128^2), the benefits are more incremental.
>
> We see no fundamental reason why learned latent representations must lose important details at coarser rollout resolutions, so long as the number of latent channels is sufficient and the original PDE fields $X_t$ exist on a manifold/attractor of sufficiently low dimension in their original ambient space. Our results show that--at least for the tasks we investigated--HREs can indeed operate efficiently and accurately at coarse resolutions, while nonetheless producing accurate outputs at high resolution (Fig. 5, 21-22). Our KS experiments (Fig. 15) also show that far from being useless, a rollout at resolution 32 can accurately predict future states at resolution 2048, but only if learned representations are used (compare HRE and mUnet performance). Overall, the key requirement seems to be a high ratio between the rollout and input resolutions rather than a low rollout resolution per se, but further experiments should investigate this more systematically.
>
> ULSEs can also be considered an extreme case of spatial coarsening with 1x1 rollout resolution (excluding the channel dimension), but clearly these can produce useful and accurate outputs at high resolution.
>
> We agree that HREs' dependence on rollout resolution, input resolution, and domain size bears further investigation. Because our models are convolutional, we can apply them to larger physical domains at the same resolutions, e.g. $1024^2$ input / $64^2$ rollouts vs. $512^2$ /$32^2$. While time, space and computational constraints prevent us from fully answering these questions in the present study, we now better emphasize them in our discussion.
>
> > For Figure 5 (experiments on high-res data), how is the error calculated for Factformer-256 if the training data is 512x512?
>
> We calculate RMSE for FactFormer-256 after coarsening reference fields to $256^2$. This essentially gives FactFormer-256 an unfair advantage over other methods whose predictions were evaluated at the higher resolution of $512^2$, though its results are still not competitive with HREs or with mUnet + superresolution (Fig. 5). We now explicitly describe how RMSE was computed for this model.
>
> > Many engineering simulations are mesh-based, which wouldn’t really admit a good averaging operator or perhaps even need coarsening. Some emulators can work with a full-resolution field/mesh, which would theoretically be better than an HRE approach.
>
> Indeed, state-of-the-art weather forecasting networks use spherical mesh representations of atmospheric states [2-4]. These methods use coarse-resolution internal representations of original mesh, which are computed using well-established procedures for graph-based message passing between meshes of different resolutions. However, unlike our HREs, mesh-based weather-prediction models have not yet included coarsened input fields, hard physical constraints, or multistep rollouts within a latent space. Similarly, UPT [5] and related methods use a cross-attention-based perceiver layer to drastically decrease the number of tokens relative to the input dimensionality. Given the ability of these "coarsening-like" operators for meshes and point clouds, we see no fundamental technical obstacles preventing the implementation of HREs for these spatial discretizations. We now note this in the "Limitations and Future Work" section of our discussion.
>
> > Line 184
>
> Corrected.
>
> [1] Wandel et al. "Learning Incompressible Fluid Dynamics from Scratch-Towards Fast, Differentiable Fluid Models that Generalize." ICLR, 2021.
>
> [2] Lam, et al. "Learning skillful medium-range global weather forecasting." Science, 2023.
>
> [3] Oskarsson, et al. "Probabilistic weather forecasting with hierarchical graph neural networks." NeuRIPS, 2024.
>
> [4] Price et al. "Probabilistic weather forecasting with machine learning." Nature, 2025.
>
> [5] Alkin et al. "Universal physics transformers: A framework for efficiently scaling neural operators." NeuRIPS, 2024.

---

> ### Comment · Reviewer_6mAC · 2025-08-01
> **Thank you for the response**
>
> Thank you for the additional experiments and for replying to my questions. I will preface this by saying that I am now inclined to recommend acceptance (it seems that authors can't see updated scores until they are officially released?). For the sake of transparency, I will give some thoughts as a reader that may help improve the quality of the work.
>
> - I apologize for my misleading terminology, I was confused between the terms rollout resolution, target resolution, and encoding resolution. I agree that 32x32 is plenty of resolution to rollout in a latent space. My intention was to ask about the target resolution, it seems most of the results the main body are run with 32x32 or 64x64 target resolution? Or maybe some higher resolution along the cost-accuracy frontier? Perhaps I am still misunderstanding, but I tried to comb through the work to figure out at what resolution the targets of most of the results/figures (1d, 2, 3) are evaluated at. The intention of my remark is that practitioners may not have much use for targets that are so coarse (32x32), if this is what the targets are being evaluated at.
> - I sympathize with the page requirement but somehow I had a hard time figuring out what experimental setup models are being compared to in the figures/results. HREs/Unets/Factformer have two resolutions they operate at (target/rollout), DINo has the target and encoding resolution, DilResnet/FNO only uses the target resolution, and above all this there is the reference resolution. It would be helpful to know these numbers for experiments in the main body. Model trends are not even consistent across these resolutions. From what I can gather from Figure 1d, when decreasing the target resolution (or faster inference speed) model performance suffers, and sometimes when increasing target resolution (or slower inference speed) model performance also suffers. As a reader it is sometimes difficult to figure out what model configurations are being compared to each other as well as the overall message of the work.
> - I am not as familiar with DINo, but at least for FactFormer, it is comparable, and sometimes worse than DilResNet and FNO, as reported in Tables 1-3 in the original work. I am somewhat surprised at how fast/inaccurate DilResNet ranks in Figure 1d, since from personal experience and some prior works (even in the FactFormer paper), it seems to be an accurate, yet slow method. What are your thoughts on this?
> - The requested experiments (LatRoll) are great and verify the effectiveness of the method. Thank you.
>
> Overall, nice job on the work and best of luck!

---

> > ### Author Response · Authors · 2025-08-09
> >
> > > Thank you for the additional experiments ...
> >
> > We thank the reviewer for the fruitful exchange and score increase (which we indeed cannot see), and for calling attention to this important but overlooked baseline.
> >
> > > I apologize for my misleading terminology ... targets are being evaluated at.
> >
> > We appreciate this point and wish to make two points in response:
> >
> > 1.**Clearer indication of experiments and their relation to figures/tables**  To track all benchmarks and baselines we added a table of experiments showing rollout and target resolutions, baseline architectures, and tasks to the appendix, in which the experiments for each figure or table are indicated. We also revised descriptions in the main text, appendix and captions giving relevant resolutions. Supplementary figures now specify resolutions in their captions.
> >
> > Most experiments use $32^2$ or $64^2$ target resolutions, with some notable exceptions.
> >
> > In Fig. 2-4, all models were trained and evaluated with target resolution $32^2$ and rollout resolution $32^2$ for FRNEs and HREs. HRE used encoding resolution $512^2$, while FF and DIno used encoding resolution of $32^2$.
> >
> > In Fig. 5 and 20-22 all models are evaluated at target resolution $512^2$ (except FF-256 at $256^2$). For HRE-64, mUnet-512 and mUnet-64+SR, the number after the dash indicates rollout resolution, while for FF the number indicates the encoding resolution; this is now clarified. For experiments with a target resolution of $512^2$, only mUnet and FF are trained on *rollout resolutions* of $512^2$ (also $256^2$ for FF).
> >
> > Fig. 1D presents combined results from multiple encoding/rollout resolutions, to accurately reflect the speed-accuracy tradeoff inherent in these choices. We use correlation instead of RMSE here for better and fairer comparisons, across both emulators and over resolution choices for the same emulator.
> >
> > 2.**Usefulness of coarse-resolution rollouts**
> > We agree that predicting PDE fields at $32^2$ or $64^2$ may not be useful for all applications, even when the predictions are accurate over long rollouts. To justify our choice to use coarse target resolutions in Fig. 1-4:
> >
> > A. For better or worse, it's standard practice in the field to evaluate at coarse resolution [1, 2, 3, 4]. Since we question standard practices involving rollout/encoding/target resolutions, introduce new representations, and call attention to these issues, we chose to incorporate standard metrics to avoid the false impression that HREs only appear better due to new metrics.
> >
> > B. For practitioners requiring high-resolution predictions, fig. 5 and 20-22 show HREs+decoder are effective. Standard emulators combined with a super-resolution step also produce reasonable results. Thus, while most of our results are at coarse target resolution, we also show HREs can produce accurate rollouts at high resolution.
> >
> > C. Even when calculating the TKE spectra at the coarse resolution of $32^2$ (Fig. 4, 18-19), there is signal loss at the highest spatial frequencies. HREs, which can access input fields at encoding resolution $512^2$, suffer least from this, while some baselines show >50% attenuation compared to the reference simulation. Since even at $32^2$ most baselines cannot reconstruct high-frequency features, evaluating them at higher-resolutions than the rollout/encoding resolutions they were designed for and trained on seemed redundant, and possibly unfair.
> >
> > > I sympathize with ... in the main body.
> >
> > See previous point above (new table and additional explanation have been added).
> >
> > > Model trends are not even consistent across these resolutions ... overall message of the work.
> >
> > We agree the results in Fig. 1d cannot be summarized as a simple trend. One potential explanation for the degradation of some models at higher rollout resolutions (e.g. mUnet-512 vs. mUnet-32 or Factformer-512 vs. FactFormer-256) is that these methods are trained directly on high-resolution targets, so a conflict can arise between the desiderata of accurately predicting high-resolution features vs. accurately predicting long rollouts. In contrast HREs train the encoder+processors using the simple loss in eq. 8, and freeze them when training the decoder, in accordance with our aim of focusing on long-rollout accuracy. We have added explicit discussion of these issues to the main text.
> >
> > contd.

---

> > > ### Author Response · Authors · 2025-08-09
> > >
> > > > I am not as familiar with DINo ...
> > >
> > > DINo uses smoother and smaller input fields for compressing data to an unstructured latent space; therefore, we believe it is normal for DINo to struggle with our KF task in particular, as the evolution of this chaotic system is highly sensitive to initial conditions.
> > >
> > > For short-term performance, we observe that DilResNet trained with added input noise is competitive. We follow the same noise variance of 1e-4 for all cases as used in the original paper [3]. However, our ID task has a different maximum peak wavenumber distribution (4.2 for ours compared to 10), which slows down the coalescence of small-scale eddies into larger ones.
> > >
> > > For the KF case, we again use the same noise schedule, although the original paper does not include this benchmark. However, [4] trains DilResNet on the KF case but uses vorticity fields instead of velocity fields and does not achieve particularly good rollout accuracy/stability for this task with DilResNet (Fig. 3 of [4] for curriculum/pushforward training already shows divergence for 1.6 second rollouts), except when training with MMD loss. The FactFormer paper reports only short rollout results for these baselines.
> > >
> > > Nevertheless, we acknowledge that tuning the noise parameter for long-term stability is challenging, and training/tuning DilResNet requires substantial resources. To strengthen this baseline, given our available resources we trained DilResNet for 500 epochs instead of 300 (DilResNet-NT below). We also trained DilResNet with no input noise and rollout curriculum of [1,2,4,8,16] with 100 epochs each for ID and KF tasks at rollout resolutions $32^2$ and $64^2$ (DilResNet-UT).
> > >
> > > These changes improved DilResNet's RMSE and correlation on ID for longer rollouts, particularly in terms of RMSE values and correlation metrics. For ID $32^2$, DilResNet now matches Factformer (whose correlation was 0.79 and RMSE was 0.66). We see more benefits from more epochs of noise training for ID $64^2$ case vs. unrolled training. For KF, both longer training and unrolled training improve the performance significantly and bring the model's performance closer to FactFormer's (whose correlation was 0.28 and RMSE was 1.12). However, for KF $64^2$, neither DilResNet variant can match the mUnet baseline.
> > >
> > > We added DilResNet-UT to our cost-accuracy scatter plot and replace the DilResNet trained for 300 epochs with the 500 epoch results (modifying other figures/tables as appropriate). These changes to training procedures do not affect inference time.
> > >
> > > **HREs vs DilResNet baselines for KF u field**
> > >
> > > | KF | Tsteps=16|  | Tsteps=32 |  | Tsteps=64 |  |
> > > |--|----|----|----|----|----|----|
> > > |  | RMSE ↓ | Corr. ↑ | RMSE ↓ | Corr. ↑ | RMSE ↓ | Corr. ↑ |
> > > | **Rollout Resolution 32 × 32** |  |  |  |  |  |  |
> > > | Ours | **0.03**±.01 | **0.99**±.00 | **0.13**±.04 | **0.99**±.01 | **0.71**±.10 | **0.69**±.27 |
> > > | DilResNet-NT | 0.23±.09 | 0.97±.02 | 0.61±.19 | 0.80±.10 | 2.83±1.35 | 0.04±.09 |
> > > | DilResNet-UT | 0.22±.08 | 0.97±.02 | 0.55±.16 | 0.84±.07 | 1.20±.22 | 0.26±.22 |
> > > | DilResNet-NT 500 Epochs | 0.17±.09 | 0.98±.02 | 0.47±.17 | 0.87±.08 | 1.16±.20 | 0.26±.20 |
> > > | **Rollout Resolution 64 × 64** |  |  |  |  |  |  |
> > > | Ours | **0.02**±.01 | **0.99**±.00 | **0.06**±.03 | **0.99**±.00 | **0.43**±.17 | **0.88**±.07 |
> > > | DilResNet-NT | 0.25±.13 | 0.96±.04 | 0.85±.47 | 0.70±.20 | 150.52±164 | 0.01±.03 |
> > > | DilResNet-UT| 0.15±.07 | 0.99±.01 | 0.46±.19 | 0.89±.07 | 3.32±1.62 | 0.08±.17 |
> > > | DilResNet-NT 500 Epochs | 0.19±.09 | 0.98±.01 | 0.57±.26 | 0.82±.13 | 7.05±13.22 | 0.13±.12 |
> > >
> > > **HREs vs DilResNet baselines for ID u field**
> > > | ID | Tsteps=16|  | Tsteps=32 |  | Tsteps=64 |  |
> > > |--|----|----|----|----|----|----|
> > > |  | RMSE ↓ | Corr. ↑ | RMSE ↓ | Corr. ↑ | RMSE ↓ | Corr. ↑ |
> > > | **Rollout Resolution 32 × 32** |  |  |  |  |  |  |
> > > | Ours | **0.05**±.01 | **0.99**±.00 | **0.12**±.03 | **0.99**±.00 | **0.30**±.09 | **0.96**±.03 |
> > > | DilResNet-NT | 0.10±.03 | 0.99±.00 | 0.27±.06 | 0.96±.03 | 26.84±9.99 | 0.09±.05 |
> > > | DilResNet-UT | 0.07±.02 | 0.99±.01 | 0.18±.05 | 0.98±.02 | 0.84±.19 | 0.73±.16 |
> > > | DilResNet-NT 500 Epochs | 0.07±.02 | 0.99±.00 | 0.18±.04 | 0.98±.01 | 1.77±.71 | 0.50±.17 |
> > > | **Rollout Resolution 64 × 64** |  |  |  |  |  |  |
> > > | Ours | **0.03**±.01 | **0.99**±.00 | **0.09**±.03 | **0.99**±.00 | **0.28**±.16 | **0.93**±.12 |
> > > | DilResNet-NT | 0.25±.34 | 0.96±.09 | 2.18±3.90 | 0.69±.17 | 1710±4513.89 | -0.01±.08|
> > > | DilResNet-UT| 0.05±.01 | 0.99±.00 | 0.12±.04 | 0.99±.01 | 6.56±3.58 | 0.05±.20 |
> > > | DilResNet-NT 500 Epochs | 0.11±.10 | 0.99±.01 | 0.45±.30 | 0.91±.10 | 20.80±31.37 | 0.04±.10|
> > >
> > > [1] Lippe et al., PDE-Refiner: Achieving Accurate Long Rollouts with Neural PDE Solvers, Neurips 2023.
> > >
> > > [2] Kochkov et al., Machine learning–accelerated computational fluid dynamics, PNAS 2021.
> > >
> > > [3] Stachenfeld et al., Learned Coarse Models for Efficient Turbulence Simulation, ICLR 2021.
> > >
> > > [4] Schiff et al., DySLIM: Dynamics Stable Learning by Invariant Measure for Chaotic Systems, ICML 2024.

---

### Official Review · Reviewer_RrQR · 2025-06-30

**Clarity:** 3
**Significance:** 2
**Originality:** 2
**Rating:** 4
**Confidence:** 4

**Summary:**

The paper proposes a new paradigm for constructing neural PDE solver based on coarsened discretization, more specifically, a decoupled encoding framework with physics constraints. Numerical experiments on several turbulent systems demonstrate the effectiveness of the proposed framework.

**Questions:**

Can the decoder be trained simultaneously with the encoder part, while the processor is trained after the autoencoding model has been trained?

**Ethical Concerns:**

["NO or VERY MINOR ethics concerns only"]

**Final Justification:**

The authors have provided sufficient clarification to address my concerns. Overall, the proposed model demonstrates strong performance and could be of significant interest to the reduced-order modeling community.

**Limitations:**

* In current experiments, only problems with structured grid (i.e. uniform equispaced grid) are studied, and it remains unknown how the proposed framework performs on unstructued grid.

* (Minor) Even with the learned encoder/decoder and learned encoding c_t, the information of the state function will inevitably be lost after downsampling from a fine discretization, and sometimes the tiny scale physics are still highly important to the dynamics, e.g. global weather, 3D turbulence

**Quality:**

3

**Strengths And Weaknesses:**

Strength:

1. The method is clearly written and easy to follow.

2. The integration of physics constraints (conservation and divergence-free) to the decoder part is straightforward but effective, which can be helpful for a wide array of other relevant tasks like super-resolution.

3. The idea of combining (non-learned) coarsened field with a learned latent encoding is novel. The empirical performance of the proposed model is strong across various benchmarks.

Weakness:

1. It is not surprising that truncating higher frequency modes promotes global stability when simulating a chaotic system. RMSE might also favor "blurrier" results. It would be interesting to study the pattern (e.g. spectrum of prediction) on high-resolution simulation in both short-term and long-term.

2. The paper ignores many prior works on building reduced-order representations and does not compare with any state-of-the-art frameworks simulating PDEs on reduced-order representations, for example, [1], [2], [3], [4] have investigated simulating PDEs in a mesh-reduced space.

[1] Han X, Gao H, Pfaff T, et al. Predicting physics in mesh-reduced space with temporal attention

[2] Zhou A, Li Z, Schneier M, et al. Text2pde: Latent diffusion models for accessible physics simulation

[3] Li Z, Patil S, Ogoke F, et al. Latent neural PDE solver: A reduced-order modeling framework for partial differential equations

[4] Wang T, Wang C. Latent neural operator for solving forward and inverse pde problems

---

> ### Author Rebuttal · Authors · 2025-07-31
>
> > physics constraints...can be helpful for a wide array of other relevant tasks like super-resolution.
>
> We thank the reviewer for making this point, which had escaped our attention. Indeed, a similar approach of employing hard physical constraints can be used for super-resolution without any prediction of temporal dynamics, and we now call attention to this fact.
>
> >It is not surprising that truncating higher frequency modes promotes global stability when simulating a chaotic system...
>
> We agree, but emphasize the difference in accuracy between our HREs and previous FRNEs operating at the same resolution. While both are stable, ours remain correlated to the reference solutions for significantly longer. We also note that HREs, despite operating at low resolution, do not really remove all information from higher frequency modes, as this can be encoded in the c-fields at coarse resolution. This seems to be the case, given the decoder's ability to reconstruct high-resolution fields.
>
> >RMSE might also favor "blurrier" results.
>
> We agree. This is a well-known limitation of RMSE-based losses, and more generally for deterministic prediction of chaotic systems, that the result becomes blurry when training on long prediction horizons. This is not specific to our method, and indeed, HREs maintain accuracy for longer horizons when evaluating at coarse or fine resolution than comparable baselines (Fig. 2, 5). While the challenge of probabilistic forecasting is essentially orthogonal to the problem we address with HREs, our approach could potentially be used to provide input to a probabilistic decoder (e.g. a diffusion model) and maintaining physical hard constraints while estimating the denoised target signal in a diffusion training loss. We now note this in our discussion.
>
> > The paper ignores many prior works on building reduced-order representations and does not compare with any state-of-the-art frameworks simulating PDEs on reduced-order representations, for example, [1], [2], [3], [4] have investigated simulating PDEs in a mesh-reduced space.
>
> We agree these references are relevant and now cite them all as suggested. We also carefully considered them as possible additional baseline methods for our experiments. We ultimately decided against directly including them, both due to the already high number of baselines we consider (now 9, including DPOT and the two addressing 6mAC's review), and for additional reasons specific to each method. However, some of our new and previous baselines do incorporate the ideas presented by these studies:
> * [4] represents an ULSE approach similar to DINo, which we already include.
> * [2] is a probabilistic ULSE approach with a diffusion model and conditioning on text inputs, complicating potential comparisons to our deterministic HREs and baselines. Our study's aim is to examine the potential of hybrid latent representations for long-term accuracy and stability in PDE emulation, while probabilistic forecasting addresses a different set of important concerns.
> * [1] proposes an encode-process-decode type of GNN that uses all-to-all temporal attention among the latent variables to propagate in time. Authors train the encoder-decoder and processor separately. Similar to [1], [3] uses the strategy of training an autoencoder first and freezing it to train the propagator. We now include the training mode from [1] and [3] with a frozen autoencoder operating in pure latent space (LatRoll-FAE) as an additional baseline (see response to 6mAC which includes the results for ID and KF cases for rollout resolutions 32^2 and 64^2). While this pure latent representation approach performs comparably to our model for short rollouts, it quickly decorrelates from the reference solution as rollout length increases. We attribute this to two main factors. First, when high-frequency features are present in the data, the lower-dimensional manifold learned by the autoencoder (which is trained without regard to the system's temporal evolution) may not represent the optimal embedding space for long-term emulation accuracy and stability. Second, rollout training helps the propagator correct its predictions and improve accuracy over longer time scales. However, when the system state representation consists only of learned latent variables without coarsened versions of the original PDE fields, the high-dimensional latent representation can diverge from the distribution of inputs the decoder encounters during training, reducing prediction accuracy.
>
> > Can the decoder be trained simultaneously with the encoder part, while the processor is trained after the autoencoding model has been trained?
>
> Yes, nothing prevents end-to-end training of HREs, or initial pre-training with an encoder/decoder pair. However, we chose to train the processor and encoder without a decoder, both to benefit from the efficient loss formulation at coarse resolution (eq. 8), and since we wanted to prioritize long-rollout performance over reconstruction of high-resolution features (see L157). We did experiment with alternative training modes (see L136-7), which we found to be less effective, and now describe these possibilities in greater detail, emphasizing there are no technical obstacles to employing these additional loss formulations and training modes with HREs.
>
> > In current experiments, only problems with structured grid (i.e. uniform equispaced grid) are studied, and it remains unknown how the proposed framework performs on unstructured grid.
>
> We agree that extension of this approach to unstructured meshes would be promising, and nothing in our approach is incompatible with message passing or (cross-) attention on graphs. We now acknowledge this as a limitation of our experimental results so far, and as a promising future direction.
>
> > Even with the learned encoder/decoder and learned encoding c_t, the information of the state function will inevitably be lost after downsampling from a fine discretization, and sometimes the tiny scale physics are still highly important to the dynamics, e.g. global weather, 3D turbulence.
>
> We agree that the encoder must discard significant information from the high resolution inputs $X_t$, due to the smaller number of degrees of freedom for $(x_t, c_t)$ compared to $X_t$. Nonetheless, it remains an open question whether/how much this actually has to degrade prediction accuracy. For most physical systems, the manifold or attractor of "realistic" system state trajectories will be lower than the dimension of the ambient space containing $X_t$, so it should be possible for the encoder to learn an effective representation that is more compact than the original high-resolution fields. At least for the tasks, dataset sizes, resolutions and time scales we examined, HREs seem to be the most effective approach at predicting future high-resolution fields. In comparison, emulators trained only on high- or low-resolution fields yielded inferior performance (Fig. 5).

---

> > ### Comment · Reviewer_RrQR · 2025-08-07
> >
> > I would like to thank the authors for the response and clarification. I keep my score and lean towards accept.

---

### Official Review · Reviewer_Gr4j · 2025-06-30

**Clarity:** 3
**Significance:** 2
**Originality:** 3
**Rating:** 4
**Confidence:** 4

**Summary:**

This paper introduces Hybrid Representation Emulators (HREs), a novel architecture for emulating partial differential equations (PDEs). The core idea is to augment the low-resolution physical fields typically used in neural emulators with a set of spatially-structured latent variables. These latent variables are generated by a separate encoder network that processes higher-resolution input data, aiming to capture fine-scale information that is lost during simple coarsening.

The paper demonstrates that this hybrid approach can achieve a more favorable cost-accuracy trade-off compared to several baselines, including convolutional (mUnet), Fourier-based (FNO), and other latent space models, across a range of benchmarks. The decoder uses a curl formulation for incompressibility of fluids flows.

**Questions:**

My evaluation of this paper hinges on whether the authors can convincingly argue that the HRE provides a fundamentally more robust and generalizable approach, rather than being a more complex model that overfits to an in-distribution data set.

- Can the authors provide examples showing out-of-distribution generalization? This could involve testing on a trajectory with a different Reynolds number, forcing term, or other physical variance than seen during training.

- Which physical constraints are actually used in each of the test cases? The curl is only applicable for divergence free flows, which other constraints are the authors using?

- Figure 17 shows the unconstrained HRE model producing large spikes in the L1 norm, while the mUnet baseline remains stable. This suggests that the hybrid representation may be prone to overfitting. Can the authors comment on this behavior?

Some additional minor suggestions :
- Plot legends: The dark dotted line representing the ground truth in the L1 norm plots (e.g., Fig. 2, Fig. 15) is a crucial reference but is missing from the legends.
- Decoder loss: The loss function used to train the decoder is not explicitly defined in the main text.
- L180 "an ULSE" should be "a ULSE"as in L181
- the PDF format is strange: it doesnt allow for search or text selections

**Ethical Concerns:**

["NO or VERY MINOR ethics concerns only"]

**Final Justification:**

The authors have provided additional information and clarifications. I have updated my score accordingly. I don't think it's necessarily a strong accept, but the paper makes the case for an interesting latent-space variant in the context of PDE learning.

**Limitations:**

The last section gives an outlook, but does not properly discuss limitations of the method. This should be added.

**Paper Formatting Concerns:**

No concerns.

**Quality:**

3

**Strengths And Weaknesses:**

Strengths:

-  The core concept of a hybrid, spatially-structured latent representation and is novel as far as I can tell. It directly addresses a known limitation of standard fixed-resolution emulators (loss of fine-scale information) while avoiding the scalability issues of fully unstructured latent space models.

-  The authors conduct an interesting set of experiments on multiple fluid systems (Kolmogorov flow, decaying turbulence, etc.) and a KS case. They compare their method against a set of state-of-the-art baselines, and provide a good view of its performance.

Weaknesses:

- A primary concern is that the HRE architecture, with its additional encoder and slightly larger parameter space, is essentially acting as a powerful data augmentation that overfits to the dynamics present in the training distribution. The following points are related to this.

- As detailed in Table 3 and acknowledged in the limitations, the test set is drawn from the same data-generating distribution as the training set (technically, this should be called a validation set). This setup does not measure the model's ability to generalize to out-of-distribution scenarios (e.g., different Reynolds numbers or forcing terms), which is a key goal for neural emulators. Without such testing, it is difficult to disentangle true modeling improvements from overfitting to the specific characteristics of the training distribution. The strong performance reported could be a result of the encoder learning features highly specific to the training data.

- The stability of the emulators appears to be limited by the training horizon. In Figure 2, the HRE's performance visibly degrades after approximately 32 steps. Given that the longest training curriculum uses rollouts of K=16, this suggests the model has learned stability primarily within its training horizon. This would be especially important for the 2D cases (1D KS is shown for more steps in Figure 8).

- The ablation study in Figure 17 is interesting but also raises questions. The unconstrained HRE ("Ours w/o physics") shows larger spikes in the L1 norm. In contrast, the unconstrained mUnet (the "MUNET" baseline, which is also unconstrained) is far more stable. Since both processors are based on a U-Net architecture, the addition of the latent channels lead to such dramatic spikes acts similarly to overfitting patterns. This suggests the hybrid representation may be inherently prone to overfitting.

- The "physical hard-constraints" seem to be directly based on previous work, and are not properly explained in the text. It is implied that all fluid flow test cases use divergence freeness, but while the paper initially talks about general PDEs, the final method seems to be tailored to 2D fluid flow cases. I would recommend to make clear in title, intro and intro ...

---

> ### Author Rebuttal · Authors · 2025-07-31
>
> > A primary concern is that the HRE architecture, with its additional encoder and slightly larger parameter space, is essentially acting as a powerful data augmentation that overfits to the dynamics present in the training distribution. The following points are related to this.
>
> We agree that generalization capability is an important aspect of PDE emulation. However, we disagree with two aspects of how this question is posed:
> 1. It is not necessary for an emulator or other ML model to generalize outside its training distribution to be useful. There are many cases in scientific and industrial ML applications where efficient, accurate processing of novel in-distribution inputs is precisely what is required, both for PDE emulation (e.g. choosing subtle optimizations for an airfoil design) and for other problems in ML (facial recognition under controlled conditions). Generalization is therefore indeed a "key goal," but not the only goal.
> 2. This is not really the accepted use of "overfitting." If a model performs well on held-out, in-distribution data it is not overfitted.
>
> To measure generalization capabilities of HREs,  we carried out new experiments to evaluate generalization performance. We used models trained on the ID task at rollout resolution of $32^2$. We changed the initial peak wavenumber for the log-normal distribution, which changes the distribution from which the initial conditions are sampled. Changing the initial peak wavenumber $u_\text{max}$ from the value of 4.2 used during training changes the decay rate of turbulence in the system. We rolled out for 64 timesteps and calculated the RMSE and correlation values for these modified ICS, for our HREs and for mUnet, a frequently employed FRNE baseline with strong performance across tasks. These experiments thus tested whether the advantage of HREs over baseline methods would persist when testing generalization to new ICs never seen during training.
>
> Unsurprisingly, the performance of both models began to degrade as the distribution of ICs used for testing moved further from the training distribution. However, our HRE approach continued to consistently outperform mUNet. These results show that the spatially structured latent variable fields learned by HREs do not overfit or overspecialize to the training distribution in a way that reduces performance on out-of-distribution inputs, compared to FRNEs that rely only on coarse features.
>
> **Rollout performance of Ours vs mUnet for changing IC distributions - field variable u, 64 steps**
> | ID | u_max=3|   | u_max=4.2 | | u_max=5 |  | u_max=6 |  |
> |--|----|----|----|----|----|----|----|----|
> |  | RMSE ↓ | Corr. ↑ | RMSE ↓ | Corr. ↑ | RMSE ↓ | Corr. ↑ | RMSE ↓ | Corr. ↑ |
> | Ours | **0.15**±.05 | **0.98**±.02 | **0.29**±.08 | **0.95**±.01 | **0.65**±.31 | **0.85**±.13 | **1.01**±.55 | **0.71**±.34 |
> | mUnet | 0.30±.12 | 0.90±.06 | 0.58±.34 | 0.78±.25 | 0.74±.42 | 0.75±.35 | 1.05±.68 | 0.61±.50 |
>
> We also carried these experiments out for the DilResNet baseline, yielding significantly worse results. We omit these here for brevity but will add them to the supplementary materials.
>
> We appreciate the reviewer raising this point, as we believe the inclusion of these generalization tests as a new supplementary figure improves our manuscript.
>
> >The stability of the emulators appears to be limited by the training horizon. In Figure 2, the HRE's performance visibly degrades after approximately 32 steps. Given that the longest training curriculum uses rollouts of K=16, this suggests the model has learned stability primarily within its training horizon. This would be especially important for the 2D cases (1D KS is shown for more steps in Figure 8).
>
> In general, existing neural PDE emulators can match the reference solution only for a finite time horizon, especially in chaotic systems such as KF. This is true for PDE emulators, weather prediction models, and other dynamical systems. While it would be desirable to have a method that trains on a finite time horizon but stays correlated for far longer, in the general case this remains an open problem and we do not claim to solve it. Rather, we show that HREs can continue to improve from longer training rollouts, while the benefits of these saturate early for previous approaches such as FRNEs (Fig. 4). Based on our results and many other studies, there is no reason to expect that rollouts during training can be avoided. HREs provide a way of making these rollouts computationally affordable at low resolution while still learning to extract and incorporate relevant features from high-resolution input fields.
>
> Also, while the predicted and reference solutions do eventually decorrelate for all known neural PDE emulators, this does not mean they all become unstable after this occurs. An unstable method would show a diverging RMSE (e.g. DilResNet in Fig. 2, mUnet-512 in Fig. 20), but this is not the case for HREs for any rollout lengths we tested.
>
> > The ablation study in Figure 17 is interesting but also raises questions. The unconstrained HRE ("Ours w/o physics") shows larger spikes in the L1 norm. In contrast, the unconstrained mUnet (the "MUNET" baseline, which is also unconstrained) is far more stable. Since both processors are based on a U-Net architecture, the addition of the latent channels lead to such dramatic spikes acts similarly to overfitting patterns. This suggests the hybrid representation may be inherently prone to overfitting.
>
> We agree that the unconstrained HRE exhibits these tendency L1 norm oscillations. Nonetheless, it is more accurate than unconstrained mUnet, not only RMSE and correlation, but also in the tendency L1 norm, especially after 48 steps. The fact that the HRE, with more channels and parameters, is more susceptible to these oscillations is perhaps not unexpected, but the point we wish to emphasize is that the hard physical constraints HRE can incorporate totally suppress these.
>
> We don't see a connection with overfitting, as these oscillations in the tendency L1 norm do not occur more for validation than for training data. Our new generalization tests (see above) show that HREs are not more prone to generalization failures than the baselines we aim to improve on.
>
> Overall, we agree this point bears further discussion, and now acknowledge in further detail these oscillations and the role of physical constraints in suppressing them in the "Ablations" section of our experimental results (starting L343).
>
> > The "physical hard-constraints" seem to be directly based on previous work, and are not properly explained in the text. It is implied that all fluid flow test cases use divergence freeness, but while the paper initially talks about general PDEs, the final method seems to be tailored to 2D fluid flow cases. I would recommend to make clear in title, intro and intro ...
>
> We agree and have thoroughly revised and expanded the explanations of these constraints and the way we impose them on HREs, using improved notation (see response to 7kAd). As noted above, we consider both the divergence-free constraint of incompressible flow, as well as conservation of momentum (all tasks), and of mass and energy (CD).
>
> >Plot legends: The dark dotted line representing the ground truth in the L1 norm plots (e.g., Fig. 2, Fig. 15) is a crucial reference but is missing from the legends.
>
> Revised as suggested.
>
> > Decoder loss: The loss function used to train the decoder is not explicitly defined in the main text.
>
> Revised as suggested, see response to 7kAd.
>
> > L180
>
> Corrected.
>
> > PDF format
>
> Corrected.

---

> > ### Comment · Reviewer_Gr4j · 2025-08-05
> >
> > Thanks for the additional information and clarifications. I will update my score accordingly. I don't think it's necessarily a strong accept, but the paper makes the case for an interesting latent-space variant in the context of PDE learning.

---

### Official Review · Reviewer_7kAd · 2025-07-01

**Clarity:** 2
**Significance:** 2
**Originality:** 3
**Rating:** 4
**Confidence:** 3

**Summary:**

This work introduces a novel paradigm for neural operator modeling that integrates the strengths of fixed-resolution and unstructured latent-space emulation. The central idea is to enable the enforcement of physical principles—such as conservation laws—even within the latent space. The approach is validated through comparative studies against several operator learning baselines in two-dimensional settings, demonstrating promising performance.

**Questions:**

1) Regarding the training procedure described on pp. 3–4, and specifically the loss function in Eq. (8): it would be helpful to explicitly clarify which components are being trained and which are held fixed during optimization.

2) Could the authors provide additional details on the decoder's training strategy? Is it trained independently of the loss in Eq. (8), or is there interaction between the two stages? Presenting a decoder-specific training objective in the form of an explicit loss function, akin to Eq. (8), would significantly improve transparency.

3) Further explanation is needed to understand how the decoder, as defined in Eqs. (13) or (15), ensures the conservation of physical quantities. In particular, how are the operators $\mathcal{D}_x$, $\mathcal{E}_x$ from Appendix A.3.1 chosen? How is the vector potential $a$ (Appendix A.3.2) learned, and how is it utilized in subsequent stages? How is the obtained information used to benefit the encoder? A more formal treatment, such as specifying loss terms that enforce these properties, would be highly beneficial.

4) Section 6 would benefit from a clearer specification of the PDE setups: What types of domains, boundary conditions, and initial conditions are employed?

5) Could the authors comment on the computational costs (in terms of time and memory) of HRE compared to FFRNE or ULSE in the training phase? Are the costs comparable, or is there a noticeable trade-off?

**Ethical Concerns:**

["NO or VERY MINOR ethics concerns only"]

**Final Justification:**

The authors have convincingly addressed my questions regarding architectural details and have provided additional comparisons with recent baseline models. Consequently, I have raised my score accordingly.

**Limitations:**

Yes

**Paper Formatting Concerns:**

* Please revise the formatting of the document. In its current state, formulas and references are inaccessible and cannot be searched, which significantly hampers readability. Notably, this issue is resolved in the appendix, suggesting an inconsistency that should be addressed throughout the main text.
* Additionally, please update the bibliography: several references listed only as arXiv preprints have since been published (e.g., [Alkin et al., 2024], [Gupta & Brandstetter, 2023], etc.).

**Quality:**

3

**Strengths And Weaknesses:**

Strengths:
* Novelty: The proposed hybrid latent representation combines elements of structured and unstructured modeling in a creative and original way.
* Significance: The paper engages thoroughly with recent literature, situating its contributions in the context of current trends in operator learning.
* Experimental validation: The experimental section is well-designed, with clearly documented ablation studies that provide useful insights into the method’s components.

Weaknesses:
* Clarity: Several parts of the paper would benefit from more detailed exposition, particularly regarding the algorithmic implementation. These points are elaborated upon in the questions below.
* Missing comparisons: Given recent developments in operator learning (OL), it would be valuable to compare the proposed method against foundation models for OL, e.g., [1,2], even if these are pretrained on different tasks. Can such models still capture relevant physics (e.g., in the KS case)? Furthermore, comparisons to Transformer-based methods tailored for fluid dynamics, such as [Alkin et al., 2024], are notably absent for the Navies-Stokes examples and should be addressed.

Overall, the paper presents an original idea and is generally well written. If the authors can improve clarity and provide more comprehensive comparisons to recent OL approaches, I would be inclined to revise my evaluation in favor of acceptance.

[1] Herde, Maximilian, et al. "Poseidon: Efficient foundation models for PDEs." Advances in Neural Information Processing Systems 37 (2024): 72525–72624.

[2] Hao, Zhongkai, et al. "DPOT: Auto-Regressive Denoising Operator Transformer for Large-Scale PDE Pre-Training." Proceedings of the 41st International Conference on Machine Learning, PMLR 235:17616–17635, 2024.

---

> ### Author Rebuttal · Authors · 2025-07-31
>
> >Clarity...
>
> We agree and have revised (see below).
>
> >Missing comparisons: ...
>
> To include comparisons to OL and FMs as suggested, we added DPOT on our ID and KF tasks. Starting from DPOT-S, whose 31M parameters make it comparable to our HREs, we fine-tune on either a smaller dataset (DPOT-S(FTS), 58 ICs for KF/90 for ID) or a larger one (DPOT-S(FTL), 116/180), for 500 epochs (learning rate 1e-4, other hyperparameters as in official implementation).
>
> DPOT's framework and pretraining data confer several advantages. For KF, pretraining on incompressible NS with forcing could improve short-rollout performance, while pretraining on compressible NS with decaying turbulence could aid long rollouts for ID and CD tasks (e.g., 64 steps for ID). Conditioning on 10 previous steps could compensate for coarsening.
>
> Nonetheless, our task-specific HREs outperform DPOT:
>
> **HREs vs Finetuned DPOT for KF u field**
>
> | KF | Tsteps=16|  | Tsteps=32 |  | Tsteps=64 |  |
> |--|----|----|----|----|----|----|
> |  | RMSE ↓ | Corr. ↑ | RMSE ↓ | Corr. ↑ | RMSE ↓ | Corr. ↑ |
> | **Rollout Resolution 32 × 32** |  |  |  |  |  |  |
> | Ours | **0.03**±.01 | **0.99**±.00 | **0.13**±.04 | **0.99**±.01 | **0.71**±.10 | **0.69**±.27 |
> | DPOT-S(FTL) | 0.14±.02 | 0.99±.01 | 0.88±.12 | 0.75±.06 | 1.63±.30 | 0.15±.28 |
> | DPOT-S(FTS) | 0.17±.02 | 0.99±.01 | 0.95±.16 | 0.70±.07 | 1.62±.27 | 0.10±.23 |
> | **Rollout Resolution 64 × 64** |  |  |  |  |  |  |
> | Ours | **0.02**±.01 | **0.99**±.00 | **0.06**±.03 | **0.99**±.00 | **0.43**±.17 | **0.88**±.07 |
> | DPOT-S(FTL) | 0.26±.03 | 0.99±.01 | 1.33±.14 | 0.57±.07 | 1.67±.28 | 0.0±.22 |
> | DPOT-S(FTS)| 0.27±.03 | 0.98±.01 | 1.27±.17 | 0.56±.08 | 1.60±.28 | 0.0±.11 |
>
> **HREs vs Finetuned DPOT for ID u field**
> | ID | Tsteps=16|  | Tsteps=32 |  | Tsteps=64 |  |
> |--|----|----|----|----|----|----|
> |  | RMSE ↓ | Corr. ↑ | RMSE ↓ | Corr. ↑ | RMSE ↓ | Corr. ↑ |
> | **Rollout Resolution 32 × 32** |  |  |  |  |  |  |
> | Ours | **0.05**±.01 | **0.99**±.00 | **0.12**±.03 | **0.99**±.00 | **0.30**±.09 | **0.96**±.03 |
> | DPOT-S(FTL) | 1.17±.19 | 0.97±.01 | 2.25±.39 | 0.53±.19 | 2.24±.6 | 0.17±.21 |
> | DPOT-S(FTS) | 1.13±.16 | 0.96±.02 | 2.34±.39 | 0.48±.21 | 2.31±.64 | 0.12±.20 |
> | **Rollout Resolution 64 × 64** |  |  |  |  |  |  |
> | Ours | **0.03**±.01 | **0.99**±.00 | **0.09**±.03 | **0.99**±.00 | **0.28**±.16 | **0.93**±.12 |
> | DPOT-S(FTL) | 1.15±.19 | 0.96±.02 | 2.13±.37 | 0.56±.2 | 1.9±.51 | 0.16±.22|
> | DPOT-S(FTS)| 1.13±.17 | 0.96±.02 | 2.31±.4 | 0.5±.20 | 2.09±.59 | 0.14±.22 |
>
> We revised sections 4, 7, 8 and 9 to include DPOT, and describe how OL/FMs relate to our approach. An HRE FM is possible, though training an encoder across domains, grids/meshes and variable fields would require careful design choices.
>
> >Comparisons to Transformer-based methods tailored for fluid dynamics...
>
> We agree UPT is a promising approach, but now already have 9 baselines (including DPOT and the two from 6mAC's review). Regarding transformer-based methods tailored for fluid dynamics, we note FactFormer and DPOT fit this description. Despite UPT's impressive results on Lagrangian flows and adaptive-mesh Eulerian pipe flows, it has not yet produced competitive results for Navier-Stokes on a regular grid in a closed system.
>
> >Regarding the training procedure described on pp. 3–4, and specifically the loss function in Eq. (8): it would be helpful to explicitly clarify which components are being trained and which are held fixed during optimization.
>
> While we previously stated on line 131 "We therefore jointly train the encoder and processor...", this did not clarify whether the decoder was also trained using this loss. We revised to emphasize that:
> * The decoder is not trained (or used) when minimizing $\mathcal L_c$ in eq. 8.
> * All encoder/processor parameters are optimized when minimizing $\mathcal L_c$ in eq. 8.
> * The processor and encoder are frozen when training the decoder (see below).
>
> >additional details on the decoder's training strategy...
>
> The decoder is trained independently with frozen encoder/processor (L157), for efficiency and so reconstructing high-res outputs doesn't conflict with long-rollout accuracy. We include the decoder loss:
>
> $$ \mathcal L_{\mathcal D} = \sum_{k=1}^K \left\lVert X_{t+k} - \mathcal D_c(\tilde x_{t+k}, \tilde c_{t+k}) \right\rVert_2^2 = \sum_{k=1}^K \left\lVert X_{t+k} - \mathcal D_c \circ \mathcal M_c^{k}\circ \mathcal E_c(X_t) \right\rVert_2^2$$
>
> We also include additional details in Appendix A.
>
> >Further explanation is needed to understand how the decoder, as defined in Eqs. (13) or (15), ensures the conservation of physical quantities. In particular, how are the operators $D_x, \mathcal E_x$ from Appendix A.3.1 chosen?
>
> We revised the main text as well as A.3.1 as follows:
>
> The fixed interpolation operator $\mathcal D_x$ and the form of $\mathcal B_c$ are PDE-specific, and chosen to enforce conservation laws and/or incompressibility as hard constraints.
>
> For KS and compressible flow, conservation of mass, energy and momentum are enforced by using an $r\times r$ local average as $\mathcal E_x$, and bilinear interpolation as $\mathcal D_x$. The network learns an additive correction to the updated fields $\mathcal B_\psi$, from which the local $r \times r$ average is removed (eq. 13) to produce a conservative update $\mathcal B_c$. To remove the local average, this average is first computed using $\mathcal E_x$, then removed at high resolution using $\mathcal D_x$.
>
> For incompressible flow, $\mathcal E_x$ applies face-averaging (Fig. 6) as in [1], while $\mathcal D_x$ interpolates horizontal velocities horizontally and vertical velocities vertically (Fig. 7). Both operations trivially conserve the incompressibility condition (consider the divergence of each cell in Fig. 7, upper left).
>
> >How is the vector potential (Appendix A.3.2) learned...
>
> We heavily revised the "decoder" section and A.3.1-A.3.2, adopting new notation and giving clear, explicit definitions. To summarize:
>
> Vector potentials are used only for incompressible flow, where they are simply output channels of the processor and decoder networks. These vector potentials are immediately used to compute divergence-free additive updates to the velocity fields and not used for any further computations (except backpropagation). There is no loss term for enforcing incompressibility, as this is a hard constraint (in [2], which used the same trick, loss-terms were to be less effective). Concretely, the processor outputs latent variables $\tilde c_t$ as well as a vector potential $a_t$, and we use the curl operator to compute $\tilde x_t = (\tilde u_t, \tilde v_t) = \nabla \times a_t + (\tilde u_{t-1}, \tilde v_{t-1})$. The decoder outputs only a vector potential $A_t$, from which we compute $X^\prime_t = (U_t, V_t)^\prime_t = \mathcal D_x(\tilde x_t) + \nabla \times A_t$, where $\mathcal D_x$ is fixed interpolation operator showin in Fig. 7. For periodic boundary conditions, it follows trivially that a periodic vector potential gives a zero-mean velocity update, so momentum conservation is also hard-constrained.
>
> > clearer specification of the PDE setups
>
> We added BCs, domain geometry and ICs for all tasks (in the appendix due to space constraints):
>
> L866: The spectral density of the initial conditions is sampled from the log-normal distribution that peaks at wavenumber of 7, then we generate random initial conditions from the normal distribution and filter the fields with FFT to match the spectral density.
>
> L867: We generate 128 experiments using 2048 × 2048 cells in $[0, 2\pi]\times[0, 2\pi]$ for the solver.
>
> L883: We generate the initial conditions using the same procedure as the KF case, with two differences: the peak wavenumber is chosen as 4.2, and the forcing term is set to f = 0. We generate 200 experiments using the same parameters and spatial resolutions as KF, except for these modifications.
>
> L888: Initial velocities are generated as a superposition of sinusoidal waves,
>     $\mathbf{v}(\mathbf{x}, t = 0) = \sum_{i=1}^{N} \mathbf{A}_i \sin(k_i \mathbf{x} + \phi_i)$
>     where $N=4$, $\mathbf{A}_i=\bar{v}/|k_i|$, $\bar{v}$ is max. initial velocity determined using the Mach number as $\bar{v}=c_s M$, where $c_s$ is the speed of sound. Wavenumbers are given by $k_i=2\pi i/L$, where $L$ is domain length, and the phases $\phi_i \sim \mathcal{U}[0, 2\pi]$. Density and pressure are initialized uniformly.
>
> > computational costs (in terms of time and memory) of HRE compared to FFRNE or ULSE
>
> We now report this for all models in Appendix E. HREs have similar training costs to mUnet for longer rollout training. For ID at rollout resolution of $64^2$, mUnet requires ≈7 hours to train vs. ≈9.5 for HREs on 8 A100s with batch size 16/GPU. For long rollouts, most HRE computations are at coarse resolution, but for short rollouts mUnet is considerably faster: at rollout length 1 mUnet needs ≈1 hour for 100 epochs on 4 A100s, and HREs ≈5 hours on 8 A100s. DINo trains in 7 hours on a single V100 at the same input resolution, but lacks accuracy. To summarize, HRE's superior cost-accuracy tradeoff incurs moderate training costs, which we now explicitly report.
>
> > formatting of the document
>
> We fixed compression errors. The PDF is searchable while images remain high-quality.
>
> > update the bibliography
>
> Adjusted along with others (Stachenfeld 2022, Li 2021, Wang 2020, Rahaman 2019, Lippe 2023, ...).
>
> [1] Kochkov et al. "Machine learning–accelerated computational fluid dynamics." PNAS, 2021.
>
> [2] Wandel et al. "Learning Incompressible Fluid Dynamics from Scratch-Towards Fast, Differentiable Fluid Models that Generalize." ICLR, 2021.

---

> > ### Comment · Reviewer_7kAd · 2025-08-04
> >
> > I am pleased with the author's detailed comments and clarifications, as well as the newly added experiments comparing the approach with more recent transformer-based models. The architecture is now much clearer and better situated within the context of existing work. As a result, I am inclined to raise my score.

---

### Official Review · Reviewer_kTab · 2025-07-02

**Clarity:** 3
**Significance:** 3
**Originality:** 3
**Rating:** 5
**Confidence:** 4

**Summary:**

In this paper, the authors propose a hybrid latent representation for learning based PDE emulations. The proposed representation enhances the coarsened PDE fields with structured latent variables extracted from high-resolution inputs. The hybrid representation is validated on multiple PDE emulation tasks and the experiments results show the representation can improve the skills of the PDE emulators.

**Questions:**

- How is the generalization ability of the proposed method?
- Regarding to the datasets, what are the differences between the training and test trajectories?
- How would the proposed method be applied to scenarios with more complex geometries (e.g., arbitrary geometries in real-world cases)/boundary conditions?
- The test cases shown in the paper are mainly fluid dynamics problems. How would the proposed method perform on more broad of problems e.g, deformable body simulations, fluid-solid coupling problems?

**Ethical Concerns:**

["NO or VERY MINOR ethics concerns only"]

**Final Justification:**

The authors responses address my concerns, I will keep my scores unchanged.

**Limitations:**

yes

**Quality:**

3

**Strengths And Weaknesses:**

- Strengths

The paper is well written and organized. The latent representation is an important topic for PDEs learning. In this paper, the authors discuss the existing fixed resolution and unstructured learned representations in detail with qualitative analysis. The proposed hybrid representation is a straightforward combination of these two, i.e., construct two branches at both the encoder and decoder for fixed and unstructured representations. The two branches merged in the processor for multistep rollout. In general, the idea is sound and intuitive, the experiment results show the proposed method can keep lowest RMSE and highest correlation during the long rollouts among various baselines.

- Weaknesses

The test cases are restricted to 1D and 2D cases with simple geometries. It would be more convincing if there were more large-scale/complex 3D scenarios.

---

> ### Author Rebuttal · Authors · 2025-07-31
>
> > The test cases are restricted to 1D and 2D cases with simple geometries. It would be more convincing if there were more large-scale/complex 3D scenarios.
>
> We agree this limitation is worth mentioning and have revised our discussion accordingly. We do wish to point out that our compressible Navier-Stokes example is, despite its simple boundary conditions and geometry, more complex than most PDEs considered for emulation.
>
> >How is the generalization ability of the proposed method?
>
> We have added new experiments to measure the generalization performance of HREs and baseline emulators to out-of-distribution initial conditions. These experiments show that HREs continued to outperform baselines as the distribution of initial conditions used for testing moved further away from the training distribution. Full details are given in the response to GR4j.
>
> >Regarding to the datasets, what are the differences between the training and test trajectories?
>
> Initial conditions for training and test trajectories are simply different samples from the same distributions, as is standard practice when training and evaluating neural PDE emulators, except when specifically measuring generalization performance (as we now do as well). We now clarify this point in Appendix C, which also contains full details of the IC distributions for each task.
>
> >How would the proposed method be applied to scenarios with more complex geometries (e.g., arbitrary geometries in real-world cases)/boundary conditions?
>
> The incorporation of complex geometries and boundary conditions is an important aim for neural emulation, but is essentially orthogonal to our aim of learning PDE system state representations that improve rollout stability and accuracy. Previous studies have provided boundary/geometry information as additional network inputs for PDE emulation on grids [1] or irregular meshes, or for weather and climate modeling [2-3]. The same can easily be done for HREs, with a trained encoder used to transform high-resolution boundary information into coarse-resolution inputs for the processor network. In the case of a GNN backbone architecture, [4] proposes a conservative GNN message passing method to satisfy conservation laws; combining this with suitable downsampling would be sufficient to implement HRE for problems with complex geometries. We now mention this in our discussion.
>
> [1] Wandel et al. "Learning Incompressible Fluid Dynamics from Scratch-Towards Fast, Differentiable Fluid Models that Generalize." ICLR, 2021.
>
> [2] Lam, et al. "Learning skillful medium-range global weather forecasting." Science, 2023.
>
> [3] Watt-Meyer et al. "ACE2: accurately learning subseasonal to decadal atmospheric variability and forced responses." npj Climate and Atmospheric Science, (2025).
>
> [4] Horie et al. "Graph Neural PDE Solvers with Conservation and Similarity-Equivariance." ICML, 2024.
>
> > The test cases shown in the paper are mainly fluid dynamics problems. How would the proposed method perform on more broad of problems e.g, deformable body simulations, fluid-solid coupling problems?
>
> While beyond our present scope, this is an interesting direction and there are no fundamental limitations to applying HREs outside fluid dynamics. The only real requirement is the availability of a coarsening operator that respects the desired conservation laws, and clearly this is the case for many solid-body and mixed-phase problems. In addition to the proposed examples, we note that biological or chemical systems described by kinetic rate equations coupled to advection and diffusion processes are ideal candidates, as these systems are naturally endowed with conservation laws encapsulated in the stoichiometric matrix of the rate equations. We now mention this as a possible future direction.

---

> > ### Comment · Reviewer_kTab · 2025-08-05
> > **Response**
> >
> > Thanks for the authors' detailed rebuttal, which addresses my questions. I will keep my score unchanged.

---

### Decision · Program_Chairs · 2025-09-17

**Decision:**

Accept (poster)

**Comment:**

This paper introduces a hybrid latent representation for PDE emulation. The hybrid latent representation consists of coarsened PDE fields and spatially structured latent variables (encoded by mUnet). The method is validated on multiple PDE emulation tasks and demonstrate that it can achieve a more favorable cost-accuracy trade-off compared to several baselines, including convolutional (mUnet), Fourier-based (FNO), and other latent space models.

The reviewers generally recognize novelty, significance, and the clarity of the paper. The reviewers also raised several concerns and questions. During the rebuttal, the authors addressed these concerns by including comparison with finetuned DPOT, novel IC distributions, an ablation where downsampled field is not used, etc. The reviewers are generally satisfied with the rebuttal. During the camera-ready, the authors are required to incorporate these additional experiments and suggested improvements into the camera-ready version of the paper. In addition to the reviewers' suggestions, I suggest that the authors put more experiment results (including some results in the appendix and the additional experiments in the rebuttal) into the main text of the paper, while some parts can be written in a more concise way (e.g., baseline description and task description).